# Technical note: Sublimation of frozen CsCl solutions in ESEM: Determining the number and size of salt particles relevant to sea-salt aerosols

Ľubica Vetráková[1], Vilém Neděla[1], Kamila Závacká[1], Xin Yang[2], Dominik Heger[3]

[1]Environmental Electron Microscopy Group, Institute of Scientific Instruments of the Czech Academy of Sciences, Brno, Czech Republic
[2]British Antarctic Survey, Natural Environment Research Council, Cambridge, UK
[3]Department of Chemistry, Faculty of Science, Masaryk University, Brno, Czech Republic
*Correspondence to:* Ľubica Vetráková (vetrakova@isibrno.cz) and Dominik Heger (hegerd@chemi.muni.cz)

**Abstract.** We present a novel technique that elucidates the mechanism of the formation of small aerosolizable salt particles from salty frozen samples. We demonstrated that CsCl may be a suitable probe for the sea salts due to their similar subzero properties and sublimation outcomes: CsCl substantially increased the visibility of the salt both during and after ice sublimation. Hence, we identified the factors that, during the sublimation of a frozen salty solution, are important in generating
fine salt particles as a possible source of salt aerosol. The number, size, and structure of the particles that remain after ice sublimation were investigated with respect to the concentration of the salt in the sample, the freezing method, and the sublimation temperature. The last-named aspect is evidently of primary importance for the preference of fine salt crystals over a large compact piece of salt: We showed that the formation of the small salt particles is generally restricted if the brine is liquid during the ice sublimation, i.e., at temperatures higher than the eutectic temperature ($T_{eu}$). Small salt particles that might
be a source of atmospheric aerosols were formed predominantly at the temperatures below the $T_{eu}$, and their structures strongly depended on the concentration of the salt. For example, the sublimation of those samples that exhibited less than 0.05 M often produced small aerosolizable isolated particles readily able to be windblown. Conversely, the sublimation of 0.5 M samples led to the formation of relatively stable and largely interconnected salt structures. Our findings are in good agreement with other laboratory studies which have unsuccessfully sought salt aerosols from e.g., frost flowers, at temperatures above the $T_{eu}$.
This study offers an explanation of the previously unexplored behaviour.

## 1 Introduction

Periods of low tropospheric ozone concentrations (e.g., < 10 ppbv) in springtime were repeatedly observed in the polar tropospheric boundary layer (Barrie et al., 1988; Bottenheim et al., 2002; Richter et al., 1998). These ozone depletion events are attributed to the inorganic halogens in the polar atmosphere (Abbatt et al., 2012; Richter et al., 1998). Bromine radicals
destroy ozone very efficiently; moreover, the number of reactive gaseous bromine can increase markedly during "bromine

explosions", in which every Br atom of a gaseous HOBr molecule can cause an extra bromide release from saline sea salt aerosol or other salty particles (Kaleschke et al., 2004). Thus, all saline crystals in sea ice zones are potential sources of reactive bromine: The proposed candidates include frost flowers (Kaleschke et al., 2004), first-year sea ice (Simpson et al., 2007), sea salt aerosol (SSA) from blowing snow (Frey et al., 2020; Yang et al., 2008, 2019), SSAs from open leads (Peterson et al., 2017), and the photochemistry within snowpack (Pratt et al., 2013). However, the exact sources of bromine in polar spring are still searched for (Abbatt et al., 2012; Swanson et al., 2022).

Chemical analyses of SSAs in inland Antarctica show that the aerosols are sulphate-depleted relative to sodium, which points to a sea-ice source of the SSA (Jourdain et al., 2008; Legrand et al., 2017; Rankin and Wolff, 2003; Wagenbach et al., 1998). The suggested mechanism of the sulphate-depletion of the SSA is mirabilite ($Na_2SO_4 \cdot 10H_2O$) precipitation from brine below −6.4 °C (Butler et al., 2016a, 2016b), a fractionation not plausible in sea spray particles generated directly from open oceans. The windblown snow particles of relatively low salinity were suggested to be an efficient source of the SSA (Yang et al., 2008, 2019). The proposed blowing snow mechanism relating to SSA production was supported by the Antarctic Weddell Sea data (Frey et al., 2020). Previously, highly saline frost flowers on thin sea ice had also been regarded as a significant source of the SSA (Kaleschke et al., 2004; Rankin and Wolff, 2003; Wolff et al., 2003); this was contradicted by both the wind tunnel experiment (Roscoe et al., 2011) and the *in situ* sublimation of frost flowers in an environmental scanning microscope (ESEM) (Yang et al., 2017). However, these two laboratory studies were performed in a relatively warm temperature range, between −5 to −18 °C; these values are above the eutectic temperature, $T_{eu} = -21$ °C, of a NaCl solution (Rodebush, 1918). Apart from our recent study that complements this paper (Závacká et al., 2022), we are not aware of any laboratory research on the formation of the SSA at temperatures below $T_{eu}$, which often occur in polar regions in winter and early spring; moreover, most of the sea ice is covered by saline ice crystals, including either high salinity formations such as frost flowers and the basal part of a snowpack or low salinity ices comprising the snow at the surface layer of a snowpack and on thick multiyear sea ice. The investigation of the impact exerted by the temperature and saline concentration on SSA formation is critical in understanding the implications to polar atmospheric chemistry and climate. This is because SSAs not only are a reservoir of various chemical compounds but also function as cloud condensation nuclei (e.g., O'Dowd et al., 1999) or even ice nucleating particles (DeMott et al., 2016; Wise et al., 2012).

In this study, we utilize an ESEM to visualize the process of ice sublimation *in-situ* as well as the structures of the CsCl salt particles formed upon the sublimation. We inspect the potential of the frozen solutions to generate fine salt particles that may become a source of salt aerosol when airborne. The number, size, and structure of those salt particles that remain after ice sublimation were analysed as the function of the salt concentration in the sample, the freezing method, and the sublimation temperature. Similarly, the effect of liquid sample evaporation slightly above the freezing point was inspected. This article follows on our previous paper in which the morphologies of salty frozen solutions were detailed (Vetráková et al., 2019) and complements our recent letter that briefly introduces this topic by imaging sea salt solutions in an ESEM (Závacká et al., 2022).

Using CsCl instead of sea salt elucidates the mechanism of the formation of the individual salt structures; therefore, this approach provides a missing piece of information, mediating detailed knowledge not obtainable from sea salts.

**2 Methods**

**2.1 Description of the ESEM**

The microscopic images were recorded using a non-commercial ESEM AQUASEM II constructed from a Tescan VEGA SEM at the Institute of Scientific Instruments of the Czech Academy of Sciences (Neděla, 2007). Compared to a conventional SEM, the ESEM does not require high vacuum conditions in the specimen chamber. The ESEM is suitable for the direct imaging of
wet, electrically non-conductive and electron beam-sensitive samples (Michaloudi et al., 2018) at chamber pressures up to 2,000 Pa; the samples do not require conductive coating (Schenkmayerová et al., 2014). Moreover, the device allows in-situ observation under dynamically changing environmental conditions (Neděla et al., 2020).

This ESEM facilitates the imaging of frozen samples. The samples are placed on the silicon pad of a Peltier cooling stage, which can reach temperatures down to −27 °C. The surface of the cooling stage was made from a pure, commercially available
ultra-flat silicon wafer (P-type, boron-doped, orientation (1, 0, 0); Ted Pella, Inc.). The front surface of the wafer was polished and scored by the manufacturer; we snapped the wafer along the scoring into small tiles. We provided no additional surface treatment. The silicon tile was glued to the Peltier cell with a highly thermally conductive adhesive, compatible with the low temperature and reduced gas pressure environment of the microscope. After each experiment, we cleaned the surface of the silicon pad with isopropyl alcohol; however, surface contamination due to the experimental usage of the Peltier stage is not
excluded. Due to confined space and electrical interference between the temperature sensor and the detector, we are not able to directly measure the temperature on the surface of the sample during the experiment inside the ESEM chamber. Therefore, the sample temperature is inferred from the temperature of the Peltier cooler as measured by Pt1000 temperature sensor (P1K0.161.6W.A.010, Innovative sensor technology IST AG, Switzerland) installed inside the stage below the silicon pad. The actual sample temperature was validated outside the ESEM at atmospheric conditions using a thermal camera (Flir A310)
and other Pt1000 temperature sensor frozen inside the sample that was placed on top of the Peltier stage. The bias between the temperature of the Peltier cooler and the actual sample temperature was no more than 2 °C.

The electron beam energy to enable the imaging amounted to 20 keV. The low beam current (100 pA) and short dwell time (14 µs) minimise the radiation damage and local heating of frozen samples. The YAG:Ce$^{3+}$ scintillation detector of backscattered electrons (BSE) is sensitive to signal from atoms with a large atomic number (Neděla et al., 2018); therefore,
very good material contrast between the ice and the CsCl salt is obtained.

## 2.2 CsCl as a proxy for sea salt

We observed the sublimation, evaporation, and structure of the residua of the CsCl solutions under various conditions (the concentration of the salt, the freezing method, and the sublimation temperature) in the ESEM. Although the project focused on identifying the parameters important for the formation of SSAs, the experiments utilized CsCl salt instead of sea salt of a complex composition in order to increase the visibility of the salt in the BSE detector (Figure S1). In the images, both the CsCl crystals and the brine are represented in white, the ice is usually black, and the silicon surface of the cooling stage is grey. Due to this contrast, we were able to easily distinguish liquid brine and solidified salt from the ice crystals and cooling stage and to monitor the whole process of sublimation, not only the structure of the residua. For example, we were able to assign the original location of various salty structures within frozen samples.

Besides the benefit of superior contrast in the ESEM, CsCl was chosen as a proxy for the sea salt due to the similarity in relevant physical properties with NaCl, which is by far the most abundant salt present in seawater. The $T_{eu}$ of the CsCl solution (the published values range from −24.83 to −22.3 °C (Chen et al., 2005; Cohen-Adad and Lorimer, 1991; Dubois et al., 1993; Fujiwara and Nishimoto, 1998; Gao et al., 2017; Monnin and Dubois, 1999)) is very close to that of the NaCl (−21 °C (Rodebush, 1918)). The eutectic solubilities of CsCl and NaCl are also similar (7.75 vs. 5.11 mol kg$^{-1}$ (Gao et al., 2017; Swenne, 1983)). Both salts can therefore be expected to exhibit a similar tendency to crystallize. The phase diagrams of CsCl-H$_2$O and NaCl-H$_2$O systems were presented in our previous publications (Vetráková et al., 2019; Yang et al., 2017). Others (Hullar and Anastasio, 2016) and we (Vetráková et al., 2019) have previously applied CsCl to understand the freezing of salty solutions. Seawater contains also other salts besides NaCl; these crystallize at various temperatures (e. g. ikaite and mirabilite have much higher eutectic point than NaCl) and may possibly perform certain secondary tasks that are not covered in this study. We tested the applicability of CsCl as the proxy for the sea salt by performing several analogous experiments using the proper sea salt. The overall sublimation behaviour was very similar to the one presented in the current study (Závacká et al., 2022). However, CsCl grant much better contrast (Figure S1) allowing the observation of subtle features otherwise hardly discernible. Thus, we are of opinion that CsCl can be a suitable probe for the sea salt when enhanced contrast is required, and the important factors affecting the structures of the sublimation residua and their potential to form salt aerosols revealed in this study might be applied also to the formation of natural SSAs. However, it should be borne in mind that the temperature relative to $T_{eu}$ must be considered in the comparisons; at −25 °C CsCl solution is below its eutectic point, whereas the eutectic point of sea water is either at −36 °C or at −54 °C (Vancoppenolle et al., 2019).

## 2.3 Freezing and sublimation

CsCl solutions with the concentrations of 0.005, 0.05, and 0.5 M (which approximates to 0.084, 0.83, and 7.8 wt%) were prepared via dissolving appropriate amounts of CsCl in MilliQ H$_2$O. These concentrations correspond to the molar

concentrations of ca. 0.29, 2.9, and 29 psu NaCl solution, respectively. The droplets with a diameter of ~ 4 mm were frozen at atmospheric pressure via three distinct methods:

(I) Spontaneous freezing of a droplet without seeding. The droplet was deposited onto the cooling stage of the ESEM. The temperature of the cooling stage was gradually lowered from room temperature (about 23 °C) down to the desired sublimation temperature ($T_{sub}$) at a cooling rate of ~ 0.5 °C/s; the sample froze spontaneously at −10 to −12 °C in the replicate experiments. Within this article, the samples are referred to as *non-seeded*. The term spontaneous freezing was used in this text because no additional nucleation agents were introduced into the samples on purpose; it must not be mistaken for homogenous ice nucleation which can occur at temperatures as low as −40 °C (Mason, 1958). Presumably, the nucleation is still initialized heterogeneously, e.g. by dust particles or on the surface of the cooling pad.

(II) Controlled ice nucleation (seeding). The droplet was placed onto the cooling stage of the ESEM; subsequently, the stage was cooled down to −3 °C and, after the thermal equilibration, several small ice crystals were added to the edge of the sample to initiate the nucleation process. Ice started to form, we could observe the movement of the freezing front with the naked eye. Eventually, the temperature was lowered to the $T_{sub}$ at a cooling rate of ~ 0.5 °C/s. In the text, these samples are referred to as *seeded*.

(III) Freezing in liquid nitrogen (LN). A droplet of the sample was deposited on a piece of aluminium foil and immersed into the LN. The foil was used to ensure a flat bottom of the sample, guaranteeing a better thermal contact with the cooling stage of the ESEM. Then, the frozen sample was transferred onto the stage precooled to the $T_{sub}$. Within the article, these samples are denoted as *LN-frozen*. In some of the experiments, small frozen spheres (a diameter of approx. 100 to 200 µm) were prepared (by spraying a solution into the LN) to evaluate the effect of the sample size.

The specimen chamber of the ESEM was closed and evacuated. The chamber pressure was maintained at 500 Pa. The gaseous environment comprised mostly blown-in (dry) air, although a small amount of water vapour was present too due to ice sublimation. $T_{sub}$ for all the above-mentioned methods was either −20 or −25 °C; the frozen samples were held at this temperature during whole sublimation. The chosen temperatures allow us to observe the differences in the sublimation just above and just below the $T_{eu}$.

The freezing methods I and II are identical with those described in our previous paper (Vetráková et al., 2019), where the freezing rates were estimated to be $(154 \pm 13)$ mm s$^{-1}$ in the non-seeded sample (method I) and ~ 0.2 mm s$^{-1}$ in the seeded one (method II). The rate of the LN freezing (method III) was not determined. Apart from the freezing rate, the three freezing methods differ in the direction of freezing (a vector perpendicular to the progressing freezing front). Estimated directions incident to these freezing methods are outlined in Figure 1.

### 2.4 Evaporation

Droplets (~ 4 mm in diameter) of the liquid 0.005 and 0.05 M CsCl solutions were evaporated in the evacuated specimen chamber of the ESEM at the air pressure of 650 Pa and the temperature ($T_{evap}$) of 2 °C.

# 3 Results and discussion

This paper describes the ice sublimation process above and below the $T_{eu}$ and the morphology of the resulting residua of the non-seeded, seeded, and LN-frozen samples. The outcomes are compared to the residua after the evaporation of the salty water.

## 3.1 Sublimation above the $T_{eu}$

When the temperature during the sublimation reached −20 °C, which is above the $T_{eu}$, the CsCl brine was still in the liquid state; no salt crystals were observed on the surface of the ice (Figures S1a, b, S2a, c, e). Liquid brine was well visible in a form of puddles on the ice surface and as a wide borderline around the ice body. Following pattern was observed for the seeded and non-seeded samples in all of the concentrations and in the 0.005 M LN-frozen samples: The liquid brine accumulated at the edge of the sample on the cooling stage. As the sublimation proceeded, the diameter of the sample decreased, and the brine

slowly shifted towards the retreating sample edge. At some point, the brine at the edge crystallized. The frozen sample continued reducing its size due to the sublimation, and soon the edge was separated from the crystallized brine. New liquid brine kept leaking out of the sample, and the situation recurred incessantly. The crystallization sequence at the edge is illustrated in Figure 2. The amount of the crystallized brine diminished progressively towards the centre of the sublimed sample because of the preceding drainage (Figures 3-5). After the sublimation, the crystalline CsCl was arranged in the form of

concentric circles (Figure 6). The salt rings were usually thick and compact close to the periphery, but they became thinner towards the centre and, especially in the samples with the lowest salt concentration, even small isolated particles occurred at the very centre of the samples (Figure 3). There are two concurrent processes that may run during the described sequence: ice sublimation and evaporation of water from the brine. The tendency of the sample to sublimate or evaporate can be expressed in terms of saturated vapour pressures above the ice and the brine, respectively. At −20 °C, the saturated vapour pressures

above ice and brine are 103 (Wexler, 1977) and 99 Pa (Text S1). The vapour pressure above the ice at −20 °C is then slightly higher than that above the brine, and the sublimation of ice is expected to be preferred slightly more than the evaporation from the brine. Nevertheless, as the difference is small, both processes occur simultaneously. The sublimation of the ice only shrinks the sample, without disturbing the equilibrium, while the evaporation of water from the brine causes its supersaturation. The brine that is in close contact with the ice can compensate for the water loss through ice melting to sustain the equilibrium brine

concentration, and thus the brine remains liquid. However, the ice melting cannot supply the water lost by evaporation from the brine at the very edge of the sample. Therefore, only the brine at the edge of the sample, far from the ice, becomes crystalline above the $T_{eu}$. In due time, the sublimation and evaporation lead to a shrinkage of the sample and the subsequent accumulation of the brine beyond the edge, followed by crystallization of the brine; these processes continually iterate slightly closer to the centre of the sample. As a result, the pieces of crystallized salt are arranged in the observed pattern of concentric (semi)circles

(Figures 2-6); the crystalline salt firmly adheres to the silicon pad of the cooling stage and cannot be blown away in our experimental conditions. These patterns of the sublimation residua (formed above the $T_{eu}$) resemble those of the "stick-slip" mechanism observed in $TiO_2$ nanoparticles from evaporating ethanol (Moffat et al., 2009).

In contrast to the output described in the preceding paragraph, the 0.05 and 0.5 M LN-frozen samples exhibited some differences (Figures 4-5, right columns). As in the previous cases, the liquid brine accumulated at the edge of the sample and, at a certain point, started to crystallize. However, the crystallization continued all over the surface of the frozen sample, and a fibrous, shell-like structure was formed on the surface. Despite its delicate appearance, the structure retained its shape even when the ice had completely sublimed.

The obtained results seem to indicate that the formation of salt aerosols at temperatures above the $T_{eu}$ is generally improbable due to the crystallization of the liquid brine into large pieces of salt unless the salt concentration is very low, and due to the adhesion of the crystalline salt to the ground (however, we admit the base of the saline ice is different in nature and so the adhesion to the ground may vary in comparison with our experiments). These results are in accordance with our previous study where we applied sea salt instead of CsCl (Závacká et al., 2022). Therefore, we believe the observed crystallization of the salt into large chunks above the $T_{eu}$ is transferrable also to natural samples and prevents the ice to become an effective source of aerosolizable particles under these conditions. Still, a portion of fine salt particles (with the size of few micrometres) may form above the $T_{eu}$ due to the brine leakage beyond the original sample position; this process is described in chapter 3.3.

### 3.2 Sublimation below the $T_{eu}$

For the sublimation to proceed below the $T_{eu}$, the temperature in our experiments was set to −25 °C; due to limitation of the cooling stage, we were not able to achieve lower temperature. Although the temperature of the surface may slightly deviate from that of the cooling stage (the sample temperature estimated with an external Pt1000 sensor was ~2 °C higher than the pre-set one), solidified CsCl brine was visibly present on the surfaces of the frozen samples under the pre-set conditions in all of the concentrations and freezing methods (the solidified salt on the frozen surface is nicely visible, e.g., in Figures 11 and 17a, d, g). In a few replicates, however, the surface brine was still liquid at this temperature due to closeness to the eutectic point; these results are not included in the article.

### 3.2.1 Concentrations of 0.005 and 0.05 M

The sublimation residua of the 0.005 and 0.05 M CsCl are displayed in Figures 7 and 8, respectively. The non-seeded and seeded samples of the two concentrations appear to be largely similar. Three types of residual salt structures were observed:

1. A flat rim at the location of the circumference of the frozen sample (Figures 7a, 7c, 8a, 8c); this rim supposedly formed from the brine that had leaked to the sample's edge before the $T_{eu}$ had been reached during the freezing processes. This formation mechanism is supported by the fact that, if the temperature temporarily rises above the $T_{eu}$ (and the brine liquifies) and then drops again, the salt rim becomes wider (such situation is shown in Figure S3). During these processes, surplus liquid brine was observed to leak to the edge of the sample.

2. Fine salt particles that appeared from beneath the samples and remained lying on the stage (Figures 7a, 7c, 7d, 8a, 8d, 9) as the samples were sublimating; typically, their widths and lengths were approximately 1 μm and several micrometres,

respectively. In the seeded samples, the microparticles were often arranged in straight lines (Figures 7c, 8c, 8d), while a random arrangement dominated in the non-seeded samples (Figures 7a, 8a). In the 0.05 M samples, they even interconnected into web-like structures instead of producing separated crystals (Figure 8c). Based on the adhesion of the fine particles to the silicon pad (Text S2), we suppose the particles were formed by the brine crystallizing at the bottom of the frozen sample, which is in contact with the cooling stage. The well-preserved regular lines of the small salt crystals on the surface of the silicon pad in the non-seeded and seeded droplets at the two lower concentrations enable us to grasp the original lamellar or cellular arrangement of ice and brine. The morphology of ice is closely related to particular freezing temperatures (Schremb and Tropea, 2016; Shibkov et al., 2003). The spacing of the lamellae depends on the freezing rate and solute concentration (Maus, 2007; Rohatgi and Adams, 1967; Wettlaufer, 1992).

3. Larger flakes (tens of micrometres in size) of an irregular shape appeared on the surface of the frozen samples during the sublimation (Figures 7b, 10, and 11). When the ice underneath these flakes sublimed, they detached from the surface but remained slightly fixed to the actual sample. As the frozen sample was retreating due to the sublimation, the flakes kept approaching to the centre of the sample until they fell down onto the stage (Figures 10 and 11). Thus, the flakes were scarcely found within the periphery, but their abundance increased towards the centre of the sublimed sample. A very similar behaviour was described and explained previously in the sublimation of a frozen colloidal suspension (Jambon-Puillet, 2019). Although we observed the flakes right on the ice surface, the crystallized brine had supposedly originated also from inside the frozen sample, as the sublimating ice continuously exposed deeper layers of the sample (such as the veins previously hidden in the bulk). Small flakes and aggregates were not affixed to the stage and could be readily blown away or split, possibly forming aerosol particles (Text S2). In the 0.05 M samples, large tufts of aggregated lichen-like structures, formed probably by the brine crystallizing in the veins between the ice grains, were abundant in the central part of the sublimed samples (Figures 8b, d). Having sizes of tens to hundreds of micrometres, the tufts were usually much larger than the salt flakes.

We processed the micrographs by using Mountains® software and estimated the abundance and size of the salt particles on the pad (Texts S3, S4). We detected particle number densities of approximately 3,400 and 13,000 $mm^{-2}$ in the 0.005 M non-seeded and seeded samples, and approximately 39,000 and 60,000 $mm^{-2}$ in the non-seeded and the seeded 0.05 M samples, respectively (Tables 1, S1). Although the particles were spread non-homogeneously over the pad, and the numerical estimates can be biased, the seeded samples evidently produced markedly more fine particles than the non-seeded ones at both the concentrations. Equivalent diameter was chosen to express the size of the residual particles; it represents a diameter of a disk whose area is equal to the area of the particle. The majority of the particles were less than 10 μm in the equivalent diameter (Figure 12). Moreover, the seeded samples generally produced smaller salt particles than the non-seeded samples (Figure 12, Table S2). The explanation of this behaviour may lie in the freezing directionality and rate. With the seeding, the sample freezes slowly at low degree of supercooling forming larger ice crystals (Vetráková et al., 2019). In contrast, non-seeded samples freeze faster (ca 150 mm $s^{-1}$) and directionally from the bottom up; such a process results in larger amount of salt on the surface. Micrographs in the figures 7 and 8 clearly show the residua in the seeded samples forming regular geometrical

patterns, indicating the salt remained mostly in the veins after freezing, whereas the non-seeded samples allowed mostly the larger particles originating in the triple junctions, pools and on the ice surface. Regarding larger number of the particles and their smaller size, seeding favours the formation of aerosolable salt particles significantly more than spontaneous freezing.

Compared to both the non-seeded and the seeded samples, the structures of the sublimation residua from the LN-frozen sample were completely different (Figures 7e, 7f, 8e, 8f), forming large fluffy tufts of salt having a lichen-like appearance. The smaller aggregates exhibited a size of about 100 μm, but the large ones exceeded the dimension of our field of view, which is about 400 μm. Unlike the other two freezing methods, the LN technique did not produce any salt rim on the original circumference of the sample, as the sample had already been much below the $T_{eu}$ when transferred onto the cold stage of the ESEM. We did

not find any smaller (< 100 μm) crystals in the sublimation residua of the LN-frozen samples. This effect is supposedly due to the fact that during the LN-freezing the sample freezes from the outside, and the freeze-concentrated brine is expelled by the growing ice towards the interior of the sample. Such sub-surface brine inclusions were revealed in LN-frozen spheres of aqueous solutions already previously (Vetráková et al., 2019). As an excess of the salt forms inside the sample, and the ice crystals emanated during the fast freezing are small, the veins between the ice crystals in the frozen sample may be more

interconnected, and thus the resulting sublimation residua form structures larger than those delivered by the other two freezing methods.

### 3.2.2 Concentration of 0.5 M

High solute concentrations yield a large amount of the brine, which is sufficiently interconnected throughout the 0.5 M samples; such a scenario then leads to the formation of relatively stable and large self-supporting structures that differed

significantly from those revealed at lower concentrations. All of the sublimation residua retained the original shapes of the frozen samples, forming salty casts in which the solidified brine delineates the shapes of the ice crystals (Figure 13). The freezing method was a major factor determining the structures of the sublimation residua. The variation of the outcomes of the three freezing methods is striking: The surface of the sublimed non-seeded sample was covered with a fibrous film of salt, through which hollows of various sizes were visible (Figure 13a); however, most of the structure underneath the film remained

hidden. This effect corresponds to the results of our previous studies (Vetráková et al., 2019, 2020) in terms of the accumulation of the solutes on the surface of the non-seeded sample, as in a globally supercooled sample cooled from the bottom the freezing likely proceeds from the bottom upwards (Figure 1a). The observed surface "skin" restricts to a large extent the access to the interior of the sample, thus preventing the more subtle and possibly fragile parts from being easily withdrawn by the blowing winds. The situation differed in the sublimed seeded sample (Figure 13b); the seeding likely promoted freezing downwards

from the surface, where the nucleating crystal had been placed (Figure 1b). Almost no film of salt was present on the surface, allowing the inner structure to be clearly discernible. The well-resolved fibrous ice casts formed after the ice had sublimed; these hollows in the salt matrix apparently retained the original sizes and shapes of the sublimed ice crystals that were very oblong, with strong parallel arrangement (Figure 13b). This well-preserved structure resembling the cells of a honeycomb

illustrates that not only the veins (triple junctions, i.e., places where three ice crystals meet) but also the whole grain walls were filled with an amount of brine sufficient to support the structure. In the LN-frozen samples, a sponge-like structure appeared after the sublimation of both the LN-frozen droplet (Figure 13c) and the LN-frozen microspheres (Figure 13d). The pores in the structure were numerous but much smaller than in the samples prepared via the other freezing methods, as abrupt freezing in LN induces the formation of small ice crystals. A comparison of the images in Figures 13c and 13d will reveal that the size of the sample does not determine the sizes of the ice crystals: The submillimetre spheres and millimetre-sized droplet exhibit similar dimensions of the pores in the residual structures. The structures hold together in large pieces even after the ice sublimation, with no collapse observed. By inspecting the sponge-like features in the detailed image of the LN-frozen residua provided in the bottom-left corner of Figure 13c, we established that the structure of the residua delineates only the veins between the ice crystals, not the whole grain boundaries, as in the seeded sample. Moreover, the internal structures of the sublimed LN-frozen samples appear denser than those of the seeded sample, due to the very different freezing rates and directionalities: Freezing from the outside expels the brine towards the interior of the sample, as demonstrated in Figure 1c. Although 0.5 M samples form relatively stable and large self-supporting structures that are not eligible for aerosol formation, these fine structures are expected to exhibit poor mechanical stability against external forces (including, for example, wind-cropping or physical collision with large mobile particles), and their potential breakdown could also lead to the production of small aerosol-sized particles.

### 3.3 Supercooled brine beyond the edge of a frozen sample

During our experiments, we often detected tiny brine droplets beyond the salt rim, where no observable frozen sample had been previously present. These microdroplets did not crystallize at the eutectic temperature and stayed supercooled (Figure 14a), even though the brine in the sample (in the left part of the panel in Figure 14a) had already fully solidified. Evaporation of water from the microdroplets resulted in salt crystallization and formation of a large number of (often) rectangular crystals (Figure 14b) having typical sizes of less than 20 μm (Figure S4). We infer the liquid state of the droplet-like features from their visual appearance and transformation to salt crystals during the observation. We did not use any other detection methods to prove their liquid state. Such supercooled brine microdroplets (and corresponding salt microcrystals) behind the circumference of the original sample were detected routinely in the non-seeded and the seeded samples of all tested concentrations at both temperatures and in the LN-frozen samples above the $T_{eu}$; however, they were absent from the LN-frozen samples below the $T_{eu}$. Thus, the generation of small microdroplets/crystals behind the perimeter of the visible sample was noticed in all samples whose temperature exceeded the $T_{eu}$ during the experiments (Figures 4, 5, 14): the non-seeded and the seeded samples had been loaded on the cooling stage in the liquid state and subsequently frozen, while the LN-frozen samples were treated externally and had already fully solidified before being loaded on the stage. Therefore, we assume that the observed effect relates to the interaction of the liquid brine with the silicon surface. Presumably, a small portion of a sample solution spread over the silicon pad and formed a thin film (invisible for the ESEM) due to surface wetting during the freezing

while the temperature was above the $T_{eu}$. Due to the evaporation, micrometric droplets were produced from the film; enhanced salt concentration in the microdroplets enabled their visualization in the ESEM. Further evaporation led to salt crystalization. Previously, wetting the upper molecular layers of a silicon (1, 0, 0) surface with water was simulated (Barisik and Beskok, 2013; Ozcelik et al., 2020); the researchers observed the water spread beyond the drop boundary. The nanometre-sized crystals observed centimetres away from the creeping front were considered a proof of precursor nanofilms (De Gennes, 1985; Qazi et al., 2019). Our findings not only confirm such nanofilms in CsCl solutions but also reveal their formation around salty ices at temperatures above the salts' eutectics. The brine leakage beyond the original sample position through precursor nanofilms at sub-zero centigrade temperatures can embody a viable mechanism of spreading the salts to the previously pristine snow in polar conditions. By this way micrometric salt crystals might form on a surface of ice or a rock.

## 3.4 Evaporation

We took a step further and observed the evaporation of the aqueous solution at +2 °C. Such conditions resemble the behaviour of sea water at solid supports on coastlines and grounds after the melting of sea ice (Keene et al., 2007). As the diameter of the observed drops is about 4 mm, they are much larger than the jet drops from bubbles bursting on the water surface (Spiel, 1995), which embody the most common sources of the sea aerosols. The evaporation of the liquid CsCl solution in the ESEM was very quick: The droplet evaporated before we evacuated the specimen chamber and made the ESEM ready for the imaging. The evaporated sample exhibited a wide salt ring at the periphery (Figure 15a, c) and mostly dendritic, sword-like salt crystals in the central part (Figure 15b, d). The evaporation residua resembled to some extent the sublimation ones above the $T_{eu}$. This is not surprising, because the evaporation-crystallization mechanisms of both the processes are similar (the salt in the frozen sample above $T_{eu}$ is dissolved in the liquid brine and evaporation of water from the brine induces salt crystallization).

## 3.5 Implications to polar atmosphere and environment

This study identifies the conditions at which the sublimation of salty frozen samples generates small particles of salt that may become a source of salt aerosols, presenting CsCl as a suitable probe for the sea salts in terms of their similar properties and outcomes of the freeze-drying process. The article complements our recent letter that introduces this topic by imaging sea salt solutions in an ESEM (Závacká et al., 2022). In the letter, an artificial sea ice prepared by spontaneous freezing sublimed at −16, −30, and –40 °C. The results show that the sizes and amount of the salt particles are comparable in the sea salt and CsCl sublimation residua for analogous sublimation temperatures and salt concentrations. The most prominent experimental drawback in using sea salt was the poor imaging contrast between the salt and the ice; all we could recognize were the final sublimation residua. Conversely, the liquid CsCl brine and the CsCl crystals were well visible on the surface of the ice (Figure S1), allowing us to observe the sublimation process, understand the sublimation mechanism, and detect the origin of the fine particles. We related the absence of liquid brine in the ice to the formation of the fine particles; these were released continually, as the ice sublimed at sufficiently low temperature. The experiments in here were performed in the narrow temperature range

between −20 and −25 °C, i.e., close to the eutectic temperature of CsCl, to show that it is the eutectic solidification that makes a difference. Without using contrasting CsCl, such information would not have been acquired. However, natural seawater contains diverse salts, and its eutectic temperature depends on the particular way of freezing: −54 °C for the Ringer-Nelson-Thompson pathway or −36 °C for the Gitterman pathway (Marion et al., 1999; Vancoppenolle et al., 2019). Recent experimental and modelling data support the latter as the reference equilibrium pathway (Vancoppenolle et al., 2019 and the references herein); thus, the sublimation results of the frozen CsCl solutions acquired at −25 °C are comparable to those of frozen seawater at −40 °C (Závacká et al., 2022).

The temperature of −36 °C, which ought to suffice for solidifying the brine according to this pathway, is often reached in polar winter-spring. Moreover, below −23 °C a fraction of solidified salt abruptly increases (11% at −23; 58% at −25; 85% at −33; 100% at −36 °C (Vancoppenolle et al., 2019)). We can speculate that, under the conditions of sequential crystallisation of sea salts, full solidification may not be needed to release of a low amount of SSA. Spatial separation of the crystallized salts and the remaining (low amount of) liquid brine, i.e., segregation by drainage of the brine to lower stages, might be sufficient for releasing some aerosolizable particles from the surface of the ice or snow. This speculation is based on the field experimental observations (the release of mirabilite-like and ikaite-like particles from young sea ice (Hara et al., 2017)) and the formation of fine particles from the frozen sea salt solutions at −30 °C (although the difference in sublimation at −30 and −40 °C is indicative of remaining liquid at −30 °C) (Závacká et al., 2022). However, the most effective formation of the aerosolizable particles is expected below the $T_{eu}$, or at least below the crystallization temperature of NaCl as the prevailing component of the sea salt.

The denoted molar concentrations of 0.005, 0.05, and 0.5 M CsCl are equivalent to the molar concentrations of 0.29, 2.9, and 28 psu NaCl solutions, respectively. The NaCl psu equivalents are listed for a straightaway comparison to the seawater salinity. These concentration values were applied to mimic the broad range of salinities in the environment: the lowest concentration approaches the salinity of surface snows on Arctic and Antarctic sea ices (Domine et al., 2004; Frey et al., 2020; Massom et al., 2001) or corresponds to the salinity of aged and multi-year sea ices (Notz and Worster, 2009); the middle one is comparable to that of young sea ice, whereas the highest concentration approximately equals that of sea water salinity and is characteristic of the basal layer on sea ice (Toyota et al., 2011) or in frost flowers (Douglas et al., 2012). Snow having a salinity even lower than in our experiments is found on aged sea ice and multi-year sea ice, where the salts are transported from nearby open leads, polynyas, or even the open ocean, rather than from the underlying saline ice via the upward wicking process (Domine et al., 2004). There, however, the surface melting and refreezing processes are then capable of increasing the local salinity (Vetráková et al., 2019), which will subsequently correspond to our low concentration experiments. The melting and formation of wet brine ponds can be induced either by a short period of warming or through the absorption of solar energy due to the existence of impurities such as black carbons and dusts (Kang et al., 2020; Warren, 1984). Then, depending on the climatic conditions, including the air temperature and amount of aerosols, both the spontaneous and seeded freezing can occur. We propose that the "seeding" condition is more common than the "spontaneous, or non-seeding" condition in the Arctic. There

are numerous items in the polar winter atmosphere that can materialize the seeds, e.g., ice crystals (diamond dusts), drifting or blowing snow particles, and various natural and man-made ice nucleating particles (e.g., dust, pollutants, black carbons). The artificial and natural freezing of seeded samples will differ in the temperature gradients: In the polar areas, surface snow or brine exposed to the air on sea ice usually face much colder temperatures than the bottom parts; thus, a large temperature gradient would be experienced across the sample. On the other hand, the temperature of the artificial samples is much more uniform, as the samples are thermally equilibrated prior to seeding and their overall size is very small.

We showed that the formation of the small particles is restricted or very limited if the brine is liquid during the ice sublimation, i.e., at temperatures higher than the $T_{eu}$: These are the conditions typical of the young sea ice with or without the frost flowers. The sublimation temperature appears to be the most critical parameter in this respect; the concentration and freezing method seem to affect the resulting structures of the sublimation residua above the $T_{eu}$ only marginally. In the given context, we can comprehend the fact that the previous studies did not succeed in seeking the SSA production from the frost flowers in a wind tunnel (Roscoe et al., 2011) and in an ESEM (Yang et al., 2017). In the former article, although the cold chamber air temperatures in most of the experiments were −30 °C or lower, the measured ice temperature reached −5 °C (Howard Roscoe and Eric Wolff, personal communications), and the temperature of the sampling line was about −10 °C (Roscoe et al., 2011). In the latter one, the sublimation of the brine-covered frost flowers at the temperatures of −5 and −17 °C yielded a large chunk of salt (Yang et al., 2017). In both of the studies, the temperature of the frost flowers was well above the $T_{eu}$; in that respect, the lack of aerosol-forming particles is in good agreement with the results of our study.

A large amount of small salt particles and fine structures that might act as a source of salt aerosols can readily form at temperatures below the $T_{eu}$. The structures of the sublimation residua that formed at this temperature strongly depended on the concentration: Lower concentrations produced small, isolated particles, while high ones resulted in large aggregates and self-supporting structures (Table 2, Figure 16). At ≤0.05 M solute concentrations, the amount of the brine was too small to allow the veins in a frozen sample to interconnect sufficiently, and therefore the structures of the veins broke apart in the process of ice sublimation (Figure 7) from the non-seeded and seeded samples. The LN-frozen samples behaved differently, but these conditions are not likely to represent those in the polar areas; they are relevant for extraterrestrial bodies (Fox-Powell and Cousins, 2021). At low concentrations (bellow 0.05 M) and $T_{sub}<T_{eu}$, the residua of both the non-seeded and the seeded samples appeared very similar: fine salt particles, small salt flakes, or small lichen-like tufts differed only in their number and size. Based on the appearance of the residua, we deduce that the lowest salt concentration (0.005 M) exhibits the highest proportion of fine particles over larger aggregates, and the ratio decreases with increasing concentration. From this perspective, the lowest salinity samples appear to be the best generators of small aerosolizable particles. Conversely, the absolute number of fine salt particles generated per mm$^2$ of the frozen sample substantially increased when the concentration rose from 0.005 to 0.05 M: In our experiments, the estimated number of the salt particles rose 11 and 5 times in the non-seeded and seeded samples, respectively (Tables 1, S1). However, the number of the particles then abruptly decreased at 0.5 M, as the highly concentrated sample did not yield any smaller salt particles and a self-supporting structure was formed instead. Thus, low to middle salt

concentrations (0.005 to 0.05 M) are required in order to generate small aerosol-sized particles below the $T_{eu}$, and samples with the middle salt concentration can be even more effective in producing aerosolizable particles than those having the low concentration. As we have not acquired enough reliable data on the relationship between salt concentration and the number of small particles, we do not attempt to extrapolate or estimate the number of small particles from samples having lower concentrations than the tested ones, and neither can we estimate the concentration at which the maximum number of small particles is formed.

We would like to show that CsCl serves as a good proxy for the sea salt in the performed experiments. Certainly, seawater freezing and sublimation exhibit some aspects where the nature of the individual salts is of central importance; however, the completed experiments with sea salt and CsCl exhibited multiple similarities in terms of the particle sizes and numerical densities. Based on the similar physical properties of the CsCl and NaCl salts near their eutectic points (Chapter 2.2) and the consistency of the experimental results concerning the sublimation residua of the sea salt (Závacká et al., 2022) and CsCl (this work), we are convinced that CsCl can be a suitable probe for the sea salt when an enhanced contrast is required, mediating detailed knowledge not obtainable from sea salts. We acknowledge that our system embodies merely a model of natural sea ice, but we demonstrate that there is a behaviour, previously undescribed, that is bifurcated by the eutectic temperature. Indeed, the kinetics of the freezing and the shape of the ice crystals can vary greatly to deviate from our model observations. However, as the outputs in the two "mild" freezing methods (spontaneous freezing and seeding) are largely similar under the conditions when aerosolizable particles are detected ($T_{sub} < T_{eu}$; conc. $\leq 0.05$ M), we believe that the concept presented in this study is, to some extent, applicable also for freezing in natural conditions.

### 3.6 Seeking for a source of SSA

The argument for the frost flowers being a source of the SSA was threefold: (1) Whether they are sulphate-depleted; (2) whether they break to form small particles; and, finally, (3) whether they cover enough area to be relevant. When the ice temperature ranges below −6.4°C mirabilite starts to precipitate out from the brine (Butler et al., 2016a, 2016b), and therefore the frost flowers growing out of the brine are sulphate-depleted, as seen in the observations (e.g., Rankin and Wolff, 2003). The temperature gradients above the thin ice are strong, but the temperature of the newly grown frost flowers is unlikely to range below the $T_{eu}$, except for the tips of the flowers. However, when the sea ice becomes thicker, the ice surface temperature will approach that of the air and may decrease below the $T_{eu}$, making aged frost flowers more prone to generating SSAs, on condition that the brine concentration is low enough. Nevertheless, the aged frost flowers are more likely to have been buried by the precipitation of snowfall, reducing the opportunity for the frost flowers to serve as a source of SSAs. Thus, **the frost flowers are not assumed to be an effective source of SSAs**, due to their higher temperature and very high salinity. Highly saline **young sea ice** is not a probable source of SSAs for similar reasons; however, the release of ikaite-like and mirabilite-like particles was detected also from fresh sea ice areas, which are supposed to be wet-surfaced, even though the details of the process remain unclear (Hara et al., 2017).

The stages of frost flowers' formation, growth, and erosion by winds were described by Hara et al., suggesting a release of Mg-rich salt particles from the surface snow that had covered the aged frost flowers (Hara et al., 2017). Both the recent field data (Frey et al., 2020) and the modelling (e.g., Huang and Jaeglé, 2017; Levine et al., 2014; Rhodes et al., 2017; Yang et al., 2019) have demonstrated the importance of **airborne saline snow particles** as a source of the SSA in polar regions. The **salty snow lying on the sea ice** has also the potential to generate the SSA at temperatures below the $T_{eu}$. On average, the snow

salinity is several orders of magnitude lower than that of sea water and frost flowers (Frey et al., 2020). Thus, according to the results presented herein, the snowpack is supposed to yield smaller salt particles and fine structures of salts upon sublimation at very low temperatures, and, therefore, to be potentially an effective source of the SSA. However, there might be significant differences depending on whether the snowpack lies on young or multi-year sea ice and whether there is flooding, depending on the snow thickness vs. ice thickness. Normally, young sea ice is thin and thus relatively warmer than thick ice, and a

temperature as low as the $T_{eu}$ could be difficult to reach on very thin young ice. Therefore, **the snowpack on multi-year ice is more likely to form numerous fine salt structures and SSAs** than the snow on relatively thin sea ice. This study favours (almost) **dry, low-salinity surfaces** when concerning the release of aerosolizable salt particles. Such a scenario is in accordance with the field observations, where all the low-salinity snow eroded from the dry surface of old sea ice by strong winds, whereas a large amount of the snow remained on young sea ice because of wet conditions (Hara et al., 2017). Thus, multi-year sea ice

and a low salinity snow lying on the sea ice (with typical salinities below 3 psu) make the formation of SSAs feasible especially at low temperatures, when the brine is already crystalline and the wetness of the surface remains very small. If the **salt concentration is very low**, e.g., below 0.085 psu (Závacká et al., 2022), as is the case of snow sputtered by sea-salts (Dominé et al., 2003) and snow on one-year sea ice (in the Antarctic, ~40% of such snow has a salinity of < 0.1 psu (Massom et al., 2001)), the formation of aerosolizable particles by strong wind agitation is not excluded even at temperatures above the $T_{eu}$,

because the brine is scattered throughout the snow matrix; such snow with a very low amount of liquid brine may be airborne, and its full sublimation allows the formation of fine salt particles, as there is not enough salt to coalesce to larger pieces.

It is clear that strong winds are required to lift either partially wet saline snow particles at temperature above the $T_{eu}$ from the snow/ice surfaces to the air, where salt particles can be formed through the loss of water vapour via evaporation or sublimation processes, or to directly erode these already crystallized salt particles from snow/ice surface at temperature below the $T_{eu}$.

Submicron-sized and ultrafine (<100 nm) SSAs were detected near Antarcica (Frey et al., 2020; Hara et al., 2011) and also in the central Arctic during the recent MOSAiC (The Multidisciplinary drifting Observatory for the Study of Arctic Climate expedition) field campaign. The enhancements of ultrafine SSAs observed in the central Arctic were mostly associated with blowing snow events (Gong et al., 2023), which is in good agreement with the model calculation, as the model already predicted ultrafine SSAs (diameter <200 nm) produced by blowing snow (Yang et al., 2019). We performed most of our

experiments with the resolution of 0.5 µm, which enabled us to obtain a full picture of the related processes, but the detection of submicron and ultrafine particles was hindered. In higher resolution images we occasionally observed the submicron particles but have been unable to evaluate their number and size due to a lack of statistically relevant data.

There are many other factors that may affect the structure of the residua; these include, e.g., the salt crystallographic system, activity of the brine solution, geometry of the cooling environment, and the presence of organic compounds that were evident in the form of a coating of inorganic minerals (Kirpes et al., 2019). Finally, we should also note that the formation of the sublimation residua will probably be accompanied by acidity changes during both the freezing and the sublimation (Vetráková et al., 2017), which may have major consequences as regards the heterogeneous reactivity of the aerosols thus formed (Pratt et al., 2013).

The implications of our findings to the polar atmosphere and environment are significant; however, further data (field, laboratory and modelling) are still necessary to confirm the conclusions.

## 3.7 Connection to the morphology and properties of the original frozen samples

The observed morphologies of the sublimation residua formed below the $T_{eu}$ bear important information about the internal environment of the original frozen samples. We previously studied the properties of the frozen samples from which the residua had formed (Vetráková et al., 2019). In the relevant article, the differences in the surface morphologies were well visible, but the observation of the interiors was insufficient. The brine structures visible on the ice surfaces and the sublimation residua of the spontaneously frozen samples sublimed at −25 °C are compared in Figure 17.

The sublimation below the $T_{eu}$ in the research reported herein allowed us to inspect also the sub-surface morphology by the ESEM more thoroughly. Another method for the direct visualization of solute locations was microtomography (μCT) (Hullar and Anastasio, 2016). The technique provided a 3D scan of the location of impurities in ice, but the resolution was much coarser than that achieved with the ESEM (~16 and 2 μm in the μCT vs ~500 and 50 nm in the ESEM for an overview and details, respectively). Thus, the researchers were able to visualize larger inclusions of brine in the ice, but the brine in the veins (65 to 88 wt% of the salt) was beyond the detection ability of the μCT method. The deduction of the morphology of a frozen sample from the morphology of a sublimation residuum can now supply the missing piece of the puzzle.

There appear to be several characteristic domains where impurities can accumulate within ice – on the surface, at the grain boundaries, in liquid inclusions, or within the ice crystals (Bartels-Rausch et al., 2014; Blackford et al., 2007; Dominé et al., 2003; Hullar and Anastasio, 2016; Light et al., 2009). The existence of various microenvironments in the vicinity of guest molecules within ice was also observed spectroscopically (Heger et al., 2011; Heger and Klán, 2007; Krausko et al., 2014; Ondrušková et al., 2018). The location of impurities in ice and snow is an important factor for their reactivity and potential release to the atmosphere, due to considerable variations in the accessibility of gas-phase oxidants and photons in the domains (Hullar and Anastasio, 2016 and the references within). For example, our experimental results indicate that the LN-frozen microspheres contain most of the brine inside the ice, not on the surface (Vetráková et al., 2020). It remains an open question if the molecules on the surface of the frozen microsphere are exposed to the incoming radiation or reactive gasses to a larger extent than the molecules inside and if and how the accessibility of these compartments depend on the temperature (Ray et al., 2011). One recent study found low oxygen concentration in viscous aerosol particle (Alpert et al., 2021); this principle may

apply to the FCS within the ice matrix. The varied outcome of LN freezing to seeded and non-seeded ones should be considered when relating the laboratory experiments to natural samples.

When comparing the sublimation residua formed below the $T_{eu}$ from the 0.5 M frozen samples (Figure 13), it is obvious that an abundance of the salt on the surface markedly depends on the freezing method. While the non-seeded sample is completely covered with salt, there is no excessive surface salt in the seeded sample and LN-frozen samples. This corroborates our previous

observations of the frozen (unsublimed) samples (Vetráková et al., 2019), where the amount of salt on the ice surface was related to the directionality of freezing. Higher amounts of surface salt in the frozen non-seeded sample can thus lead to enhanced oxidation of halogens by atmospheric gases and their release to the atmosphere. The sublimation of ice from the concentrated frozen samples leads to the formation of large salt chunks (above $T_{eu}$, Figure 5) or highly porous structures (below the $T_{eu}$, Figure 13). Therefore, the sublimation temperature can significantly affect the accessibility of gases and photons to the

salt and thus alter the reactivity within these compartments.

The inspection of the residua formed by the sublimation of the 0.5 M frozen samples below the $T_{eu}$ (Figure 13) also indicated substantial differences in the thickness of the brine compartments in the interior of the seeded and the LN-frozen samples. Regrettably, we could not evaluate this parameter in the non-seeded sample, as its surface was covered with a salt film to a large extent and the internal morphology was hidden. In the seeded sample, the salt crystallized in the form of relatively thick

walls surrounding individual ice crystals; similar outcome was recently observed for frozen and sublimed Enceladus ocean fluids (Fox-Powell and Cousins, 2021). In the LN-frozen sample, the salt crystallized as very thin tortuous veins – the overall shape of the residuum resembled a sponge. The most important differences between these two preparation methods consisted in the directionality and freezing rate. Seeding is a method of slow freezing from the upper surface (Figure 13b), generating large ice crystals, while LN-freezing proceeds under fast freezing from all sides (Figure 13c), generating small ice crystals

(Vetráková et al., 2019). Thus, the freezing rate and/or directionality affect the thickness of the brine compartments in a frozen sample. It is intuitively obvious that the brine surrounding small ice crystals with large specific surface areas needs to spread more, and therefore the grain boundaries and veins containing the salt ought to be thinner than in the sample of identical concentration containing larger ice crystals. In this study, this relationship is proved experimentally (Figure 13b vs. 13c). Previously, slow cooling rates were related to a larger extent of aggregation of impurities in ice, (Heger et al., 2005) while fast

cooling rates led to partial vitrification of the brine (Fox-Powell and Cousins, 2021; Imrichová et al., 2019; Ondrušková et al., 2020), presumably due to confined space and more efficient cooling of thin veins. The subtle differences in the ice-impurity morphology can possibly alter the immediate molecular environment and consequently variate the absorption (Bononi et al., 2020), aggregation (Kania et al., 2014), pH changes, (Heger et al., 2006) and, therefore, also the reactivity of compounds and their photochemical quantum yields (Hullar et al., 2018; Kahan and Donaldson, 2007; Klánová et al., 2003).

Herein we observed several characteristic units among the sublimation residua. The residua formed above the $T_{eu}$ cannot be assigned to particular domains in the ice, because liquid brine from the individual domains pours together after the ice has sublimed. However, the flow of the crystallized salt below the $T_{eu}$ is restricted, and the shape of the residua (fine particles, salt

flakes, and lichen and sponge-shaped tufts) partially mirrors the one of the domains in the ice. From our observations of the sublimation processes, we deduce the genesis of the individual salt features as follows: Fine salt particles are likely formed by crystallization of the brine in the veins at the bottom of the sample. Arguments supporting this opinion were already outlined above (adhesion of the particles to the cooling pad, their visual absence on the gradually sublimed ice surface). However, we cannot exclude that a portion of the fine particles comes from thin or separated veins inside the ice body that broke to small pieces after ice had sublimed. Remnants of the crystalline salt from larger and more interconnected veins can be presumably found in the form of lichen-like tufts. Lastly, crystallization of the brine in pools on the surface and liquid inclusions in the ice body likely led to formation of 2-dimensional salty flakes. The knowledge of the morphologies brought up in this paper and their relationship to the properties of original frozen samples should be borne in mind when designing laboratory experiments to simulate natural ice and snow. As the properties of frozen samples largely depend on the sample preparation method, artificial samples must be prepared with care in order to mimic natural ice and snow.

**5 Summary**

In this study we present a novel technique that facilitates identification of the most suitable conditions for formation of salt aerosols. We used CsCl as a proxy for natural sea salts due to its excellent visibility within the ice by the ESEM, and observed the transformation of frozen salty solutions into residual salt particles. By inspecting the morphology and number of the resulting salt particles based on the salt concentration, freezing method and sublimation temperature, we demonstrated that the sublimation temperature is the most important factor for the formation of aerosolable particles upon sublimation of various frozen salty ices. When the sublimation takes place at temperatures above the $T_{eu}$, the formation of large compact pieces of salt with very little aerosol-forming potential is preferred for all the concentrations and freezing methods tested herein; the concentration and the freezing method seem to play a less important role in the structure of the sublimation residua. Similar effect was observed when a liquid salty solution evaporated. Conversely, the structures of the residua formed at temperatures below the $T_{eu}$ strongly depend on the concentration: Low salinity samples yield small isolated particles and small aggregates (< 10 µm) that are directly available to be windblown and become salt aerosols; high salinity samples transform into large aggregates with fine structures. Regarding the size and abundance of the fine particles in various residua, the 0.05 M seeded sample yielded the largest number of fine salt particles and thus would be the most promising source of salt aerosols. The outcomes of this article clearly indicate that the sublimation process on highly saline ices, such as frost flowers lying on young sea ice, does not directly induce the formation of SSAs unless the interconnected salt structures are potentially brought down by external forces, such as physical collision with large mobile particles. On the other hand, less saline snow lying on aged ice is more likely to directly generate separated fine salt particles during the sublimation process at low temperatures, especially below the $T_{eu}$. Our data emphasize the requirement of very low temperatures for the effective formation of SSAs from the ice

and snow with the salt concentrations 0.005-0.05 M. This condition favours salty snow over frost flowers as an efficient source of the SSA, provided that no flooding has occurred.

Inspecting CsCl-containing frozen and sublimed samples in the ESEM generally led to very similar outcomes in comparison with the sea salt (Závacká et al., 2022), but provided us with many details in the samples' morphology that would otherwise remain unseen. We are aware of the limitations in using artificial samples to mimic natural processes. However, such systematic study where we vary parameters (sublimation temperature, concentration, freezing method) and monitor the outcome are very difficult (and nearly impossible) to perform with natural samples, as there are many collateral effects in

nature that we are not able to fully distinguish and interpret. Nevertheless, evaluation of these effects step by step in a laboratory may be very helpful when interpreting the behaviour of complex natural samples.

## Author contributions

Ľ.V. conducted the experiments, analysed the data, and prepared the manuscript; K.Z. performed auxiliary experiments; D.H. and X.Y. developed the idea of the project; and V.N., D.H., and X.Y. contributed to the discussions and writing of the final

paper.

## Declaration of Competing Interest

The authors declare that they have no known competing financial interests or personal relationships that could have appeared to influence the work reported in this paper.

## Acknowledgement

This work was supported by Czech Science Foundation via project GA22-25799S. The study was supported by Strategy AV21 – programme #23 "City as a laboratory of change; construction, historical heritage and place for safe and quality life". We thank Martin Olbert for the particle size evaluation and Přemysl Dohnal for the language corrections.

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

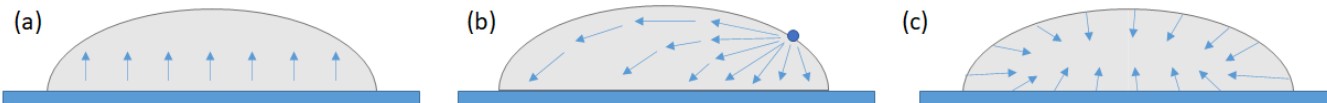

**Figure 1: The directionalities of the three freezing methods: (a) Spontaneous freezing (non-seeded sample); (b) controlled ice nucleation (seeding); and (c) freezing in LN. The arrows show the expected directions of the progress of the ice crystal growth front.**


**Figure 2: The sublimation process in the 0.5 M non-seeded sample at −20 °C (above the $T_{eu}$). The images were recorded at the edge of the sample, within intervals of 10 s. The black body in the upper part of the panels a-d represents sublimating ice, the grey background in the lower part of the panels represents a silicon surface of the cooling stage. Liquid brine is well visible in a form of puddles on the ice surface (white spots) and as a wide (white) borderline around the ice body. Farther from the sublimating ice, the salt already crystallized (bright white structures with a surface pattern). In the panels a and b, the crystallized CsCl salt is overexposed due to more intensive signal in comparison with the brine; to eliminate this, the sensitivity of the detector was subsequently lowered (panels c and d). The scale in panel *c* applies to all of the images.**

Non-seeded sample        Seeded sample        LN-frozen sample

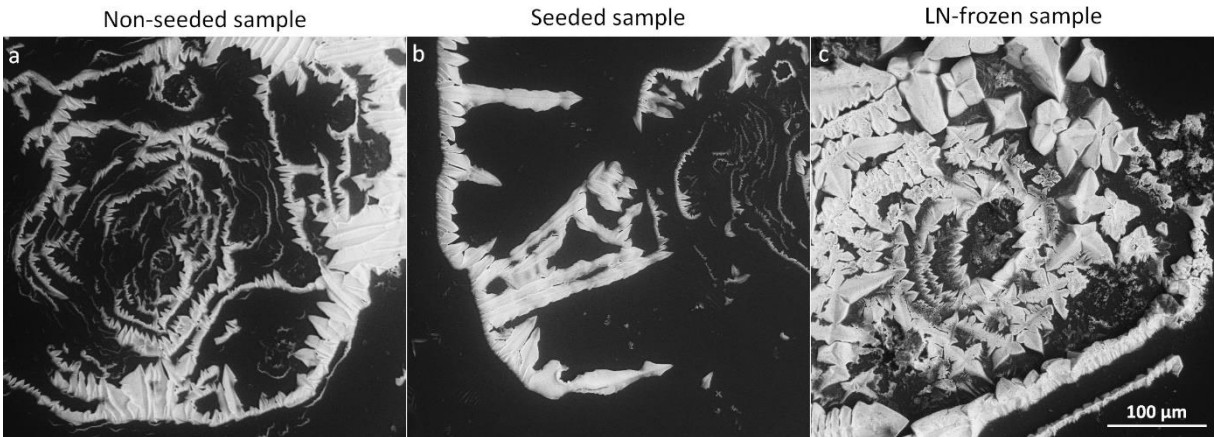

100 µm

**Figure 3: The salt residua after the sublimation of the frozen samples prepared from 0.005 M CsCl via the indicated freezing methods. The samples sublimed at −20 °C (above the $T_{eu}$). The scale in panel *c* applies to all of the images.**

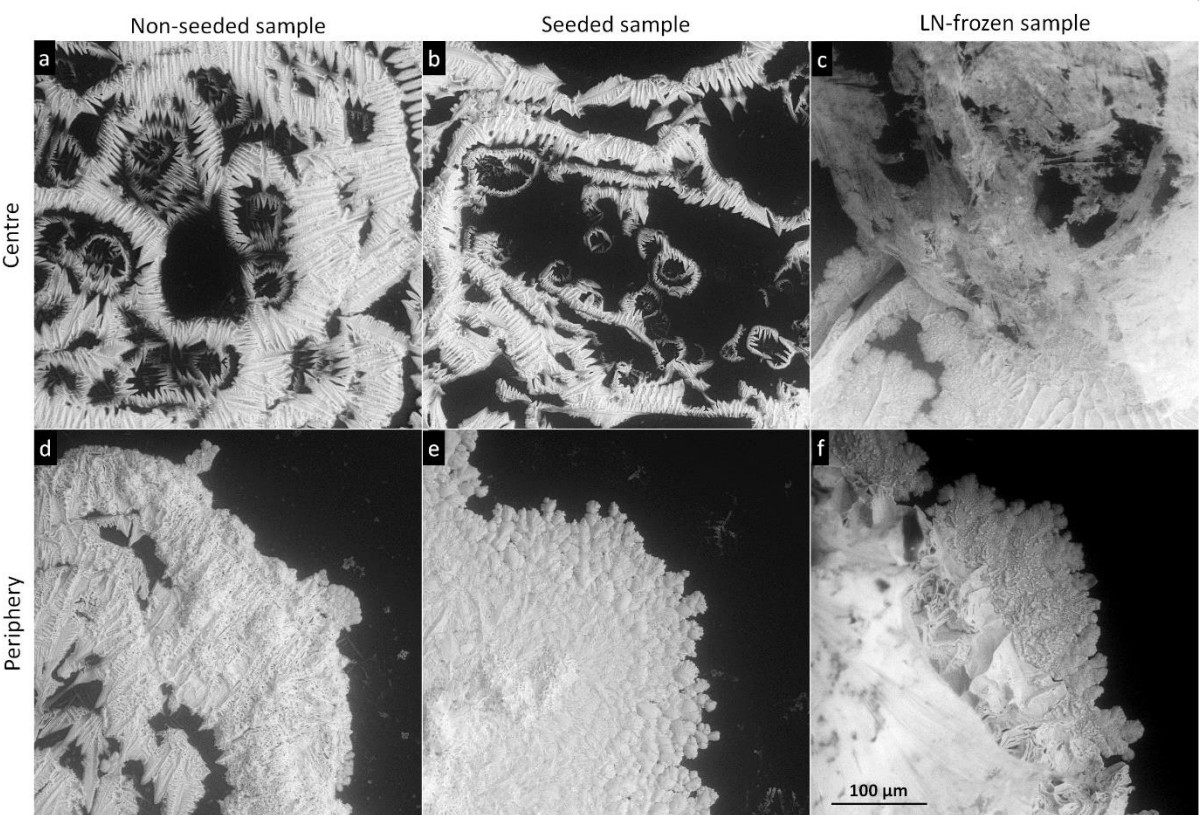

**Figure 4: The salt residua after the sublimation of the frozen samples prepared from 0.05 M CsCl via the indicated freezing methods. The centre and the periphery of the samples are displayed. The samples sublimed at −20 °C (above the $T_{eu}$). The scale in panel f applies to all of the images.**

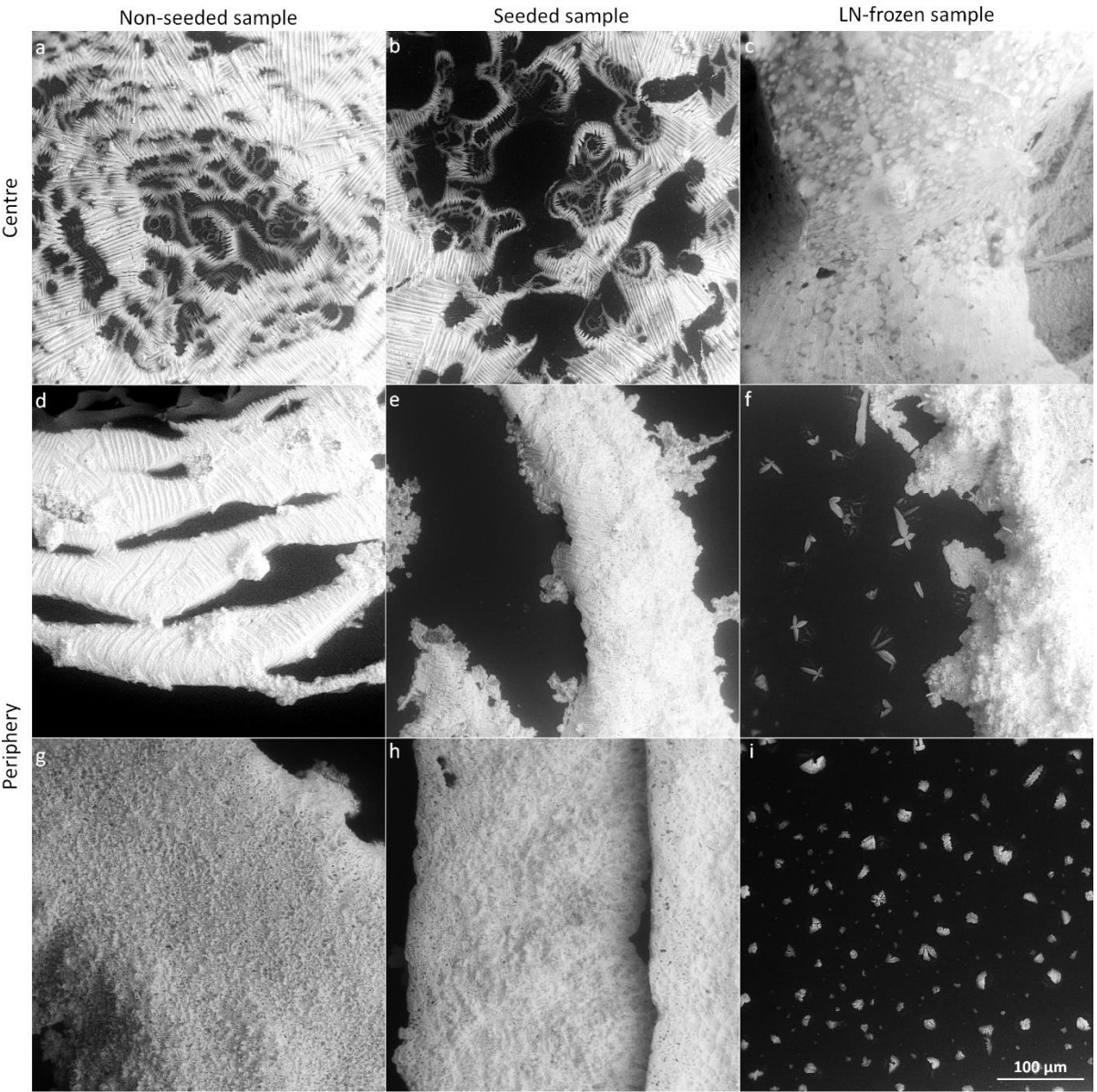

Non-seeded sample      Seeded sample      LN-frozen sample

Centre

Periphery

100 µm

**Figure 5: The salt residua after the sublimation of the frozen samples prepared from 0.5 M CsCl via the indicated freezing methods. The centre and the periphery of the samples are displayed. The samples sublimed at −20 °C (above the $T_{eu}$). The scale in panel i applies to all of the images.**

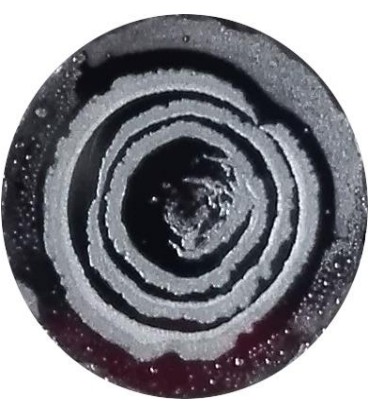

**Figure 6: A photograph of the cooling stage of the ESEM covered with the crystallized CsCl salt after the sublimation of the 0.5 M non-seeded sample at −20 °C. The diameter of the stage is about 6 mm. The pattern of the concentric circle is nicely visible.**

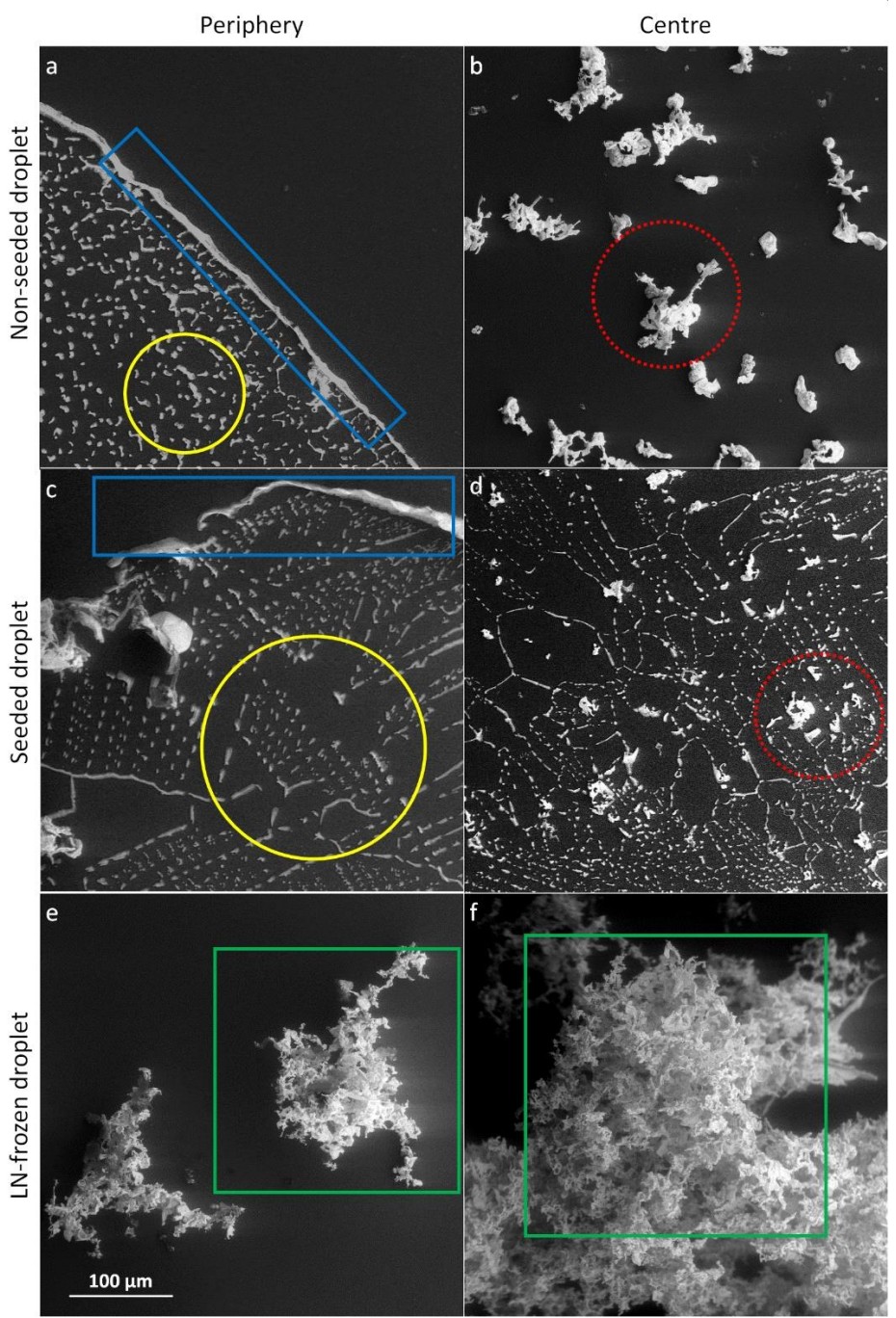

**Figure 7: The salt residua after the sublimation of the frozen samples containing 0.005 M CsCl; the sublimation temperature was −25 °C (below the $T_{eu}$). Each row represents one freezing method, named on the left-hand side. The images in the left- and right-hand columns were taken at the periphery and close to the centre of the original sample, respectively. The scale in panel *e* applies to all the images. The examples of the salt rim (blue rectangles), fine salt particles (yellow circles), salt flakes (red dotted circles), and lichen-like tufts (green squares) are highlighted.**

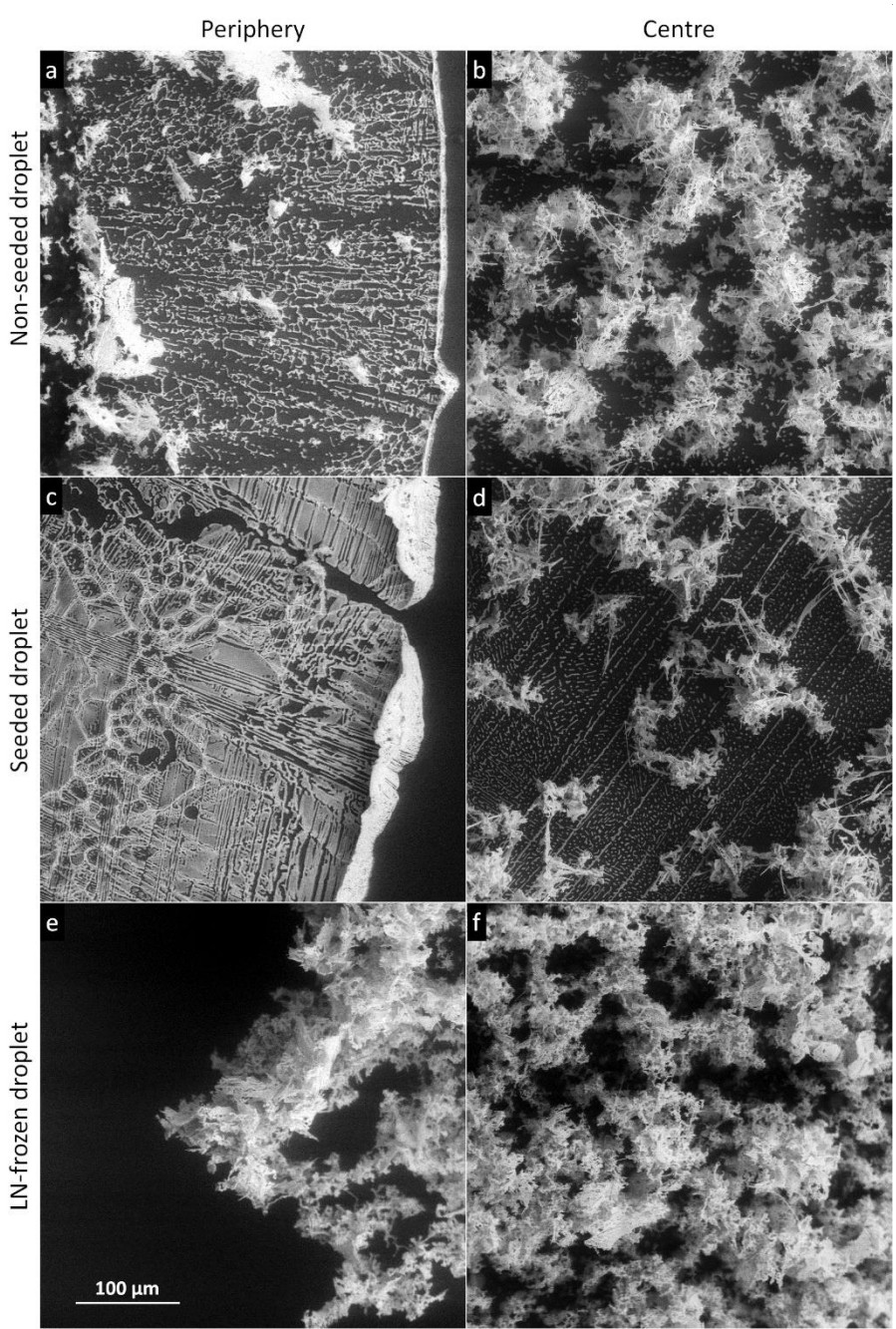

Periphery  Centre

Non-seeded droplet

Seeded droplet

LN-frozen droplet

100 μm


**Figure 8: The salt residua after the sublimation of the frozen samples containing 0.05 M CsCl; the sublimation temperature was −25 °C (below the $T_{eu}$). Each row represents one freezing method, named on the left-hand side. The images in the left- and right-hand columns were taken at the periphery and close to the centre of the original sample, respectively. The scale in panel *e* applies to all of the images.**

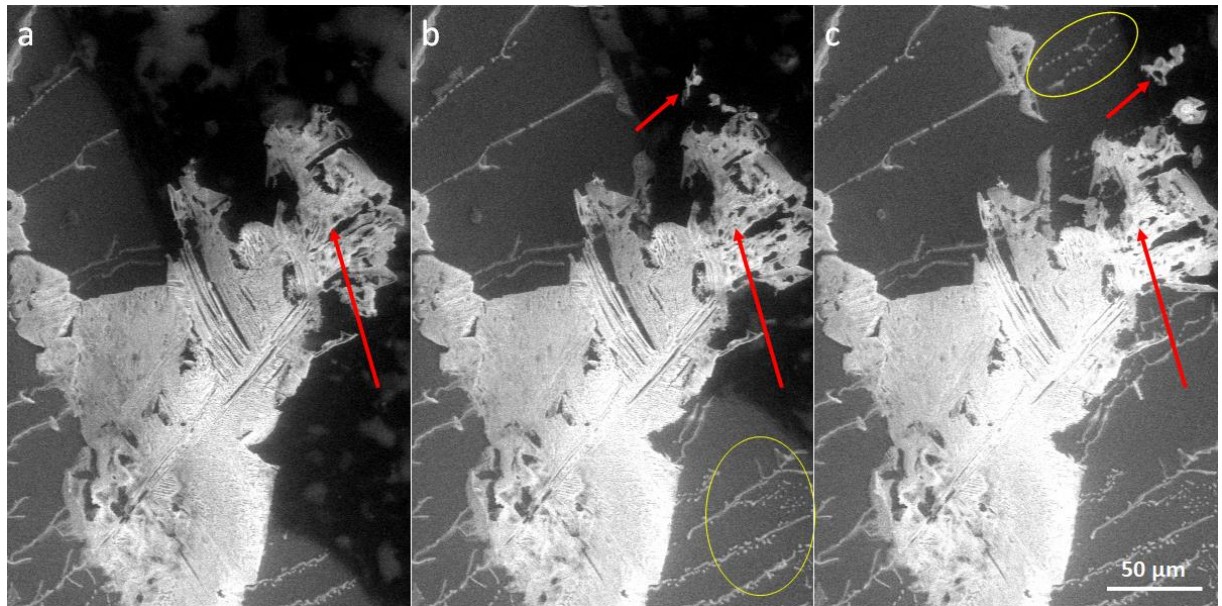


**Figure 9: The retreating ice (black) in the 0.05 M seeded sample reveals tiny salt particles (the white particles encircled in yellow) lying on the cooling stage (grey). These microparticles were most likely formed by the brine crystallizing at the bottom of the sample, in contact with the stage. Conversely, the larger salt flakes (pointed to by the red arrows), which emerge on the surface of the sample before falling down onto the stage, originate from the body and the surface of the sample.**

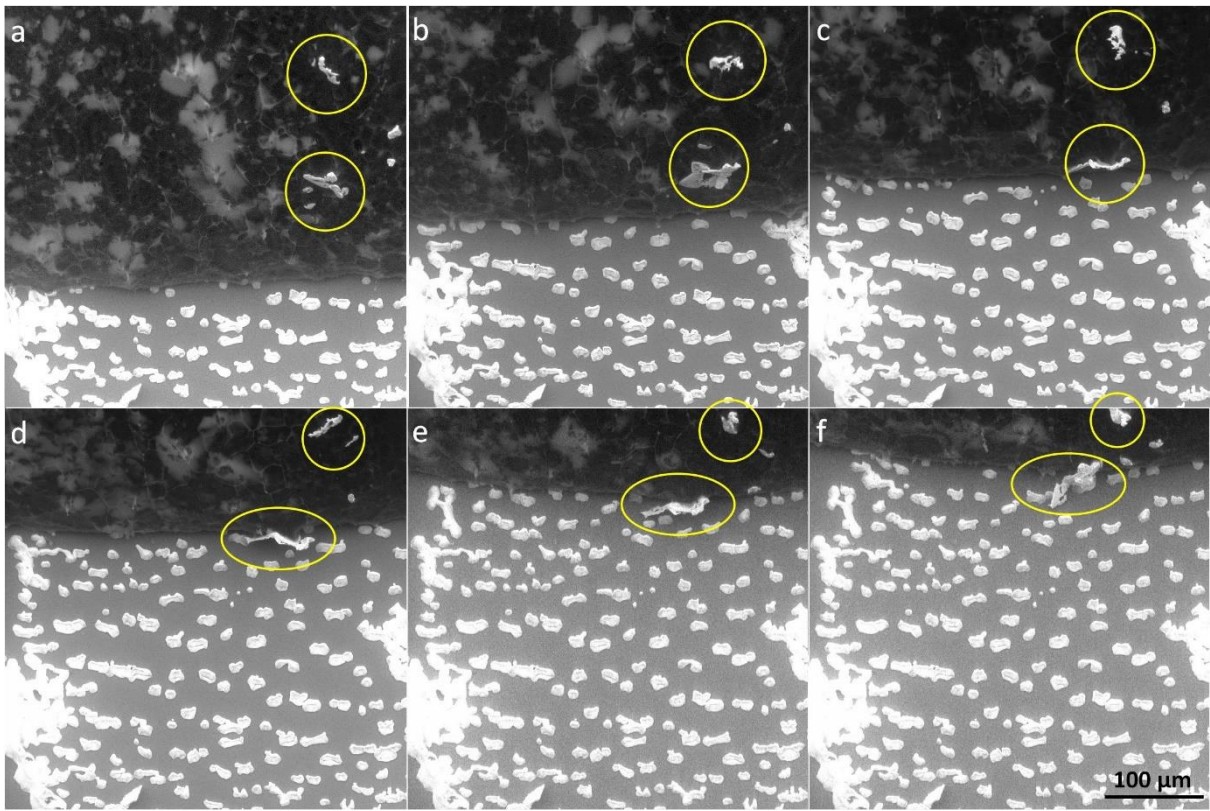

Figure 10. The retreating ice (black, upper part) in the 0.005 M non-seeded sample reveals salt particles (white) lying on the cooling stage (grey, lower part). The sublimation occurs at −25 °C. The encircled objects are the salt flakes on the ice surface.

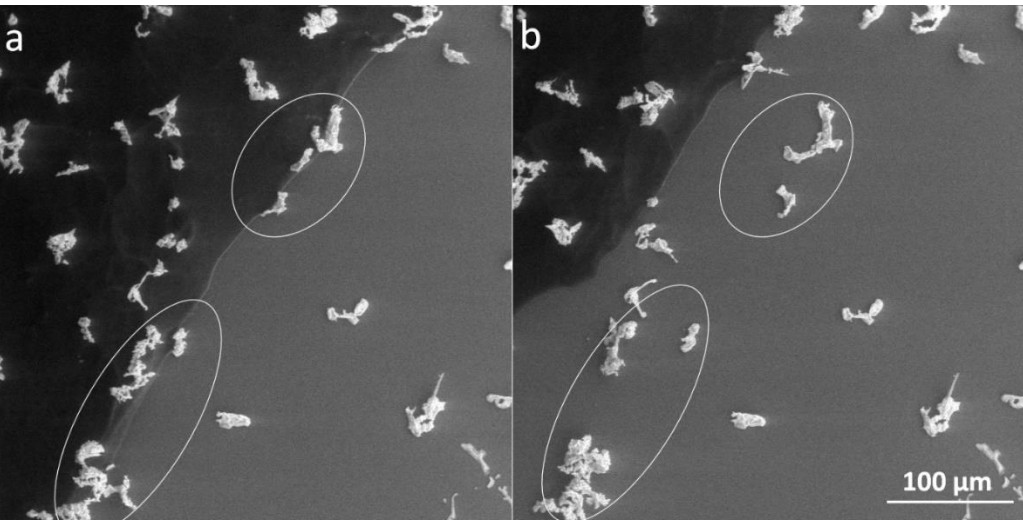

**Figure 11: The salt flakes (white) falling down the surface of the retreating ice (black, left upper part) onto the cooling stage (grey, right part) in the 0.005 M seeded sample.**

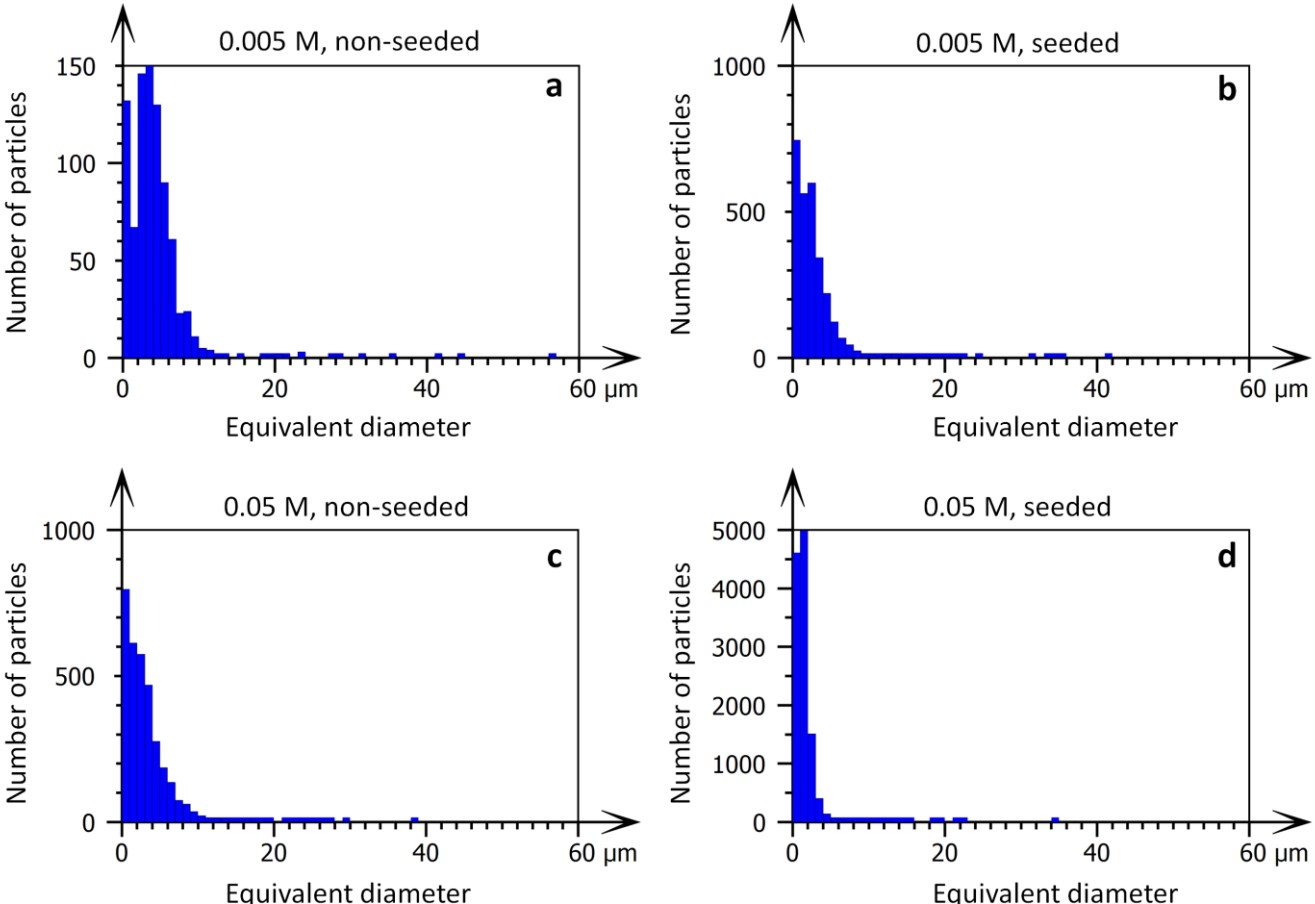

**Figure 12: The histograms showing the distribution of the equivalent diameters of the residual salt particles in the non-seeded and seeded CsCl samples sublimed at −25 °C (the equivalent diameter expresses the diameter of the disk whose area is equal to the area of the particle). The smallest applied interval is 1 μm. Each histogram embraces the particles from 3 representative images of the sublimed samples. The data were recorded with the instrumental resolution of 0.5 μm.**


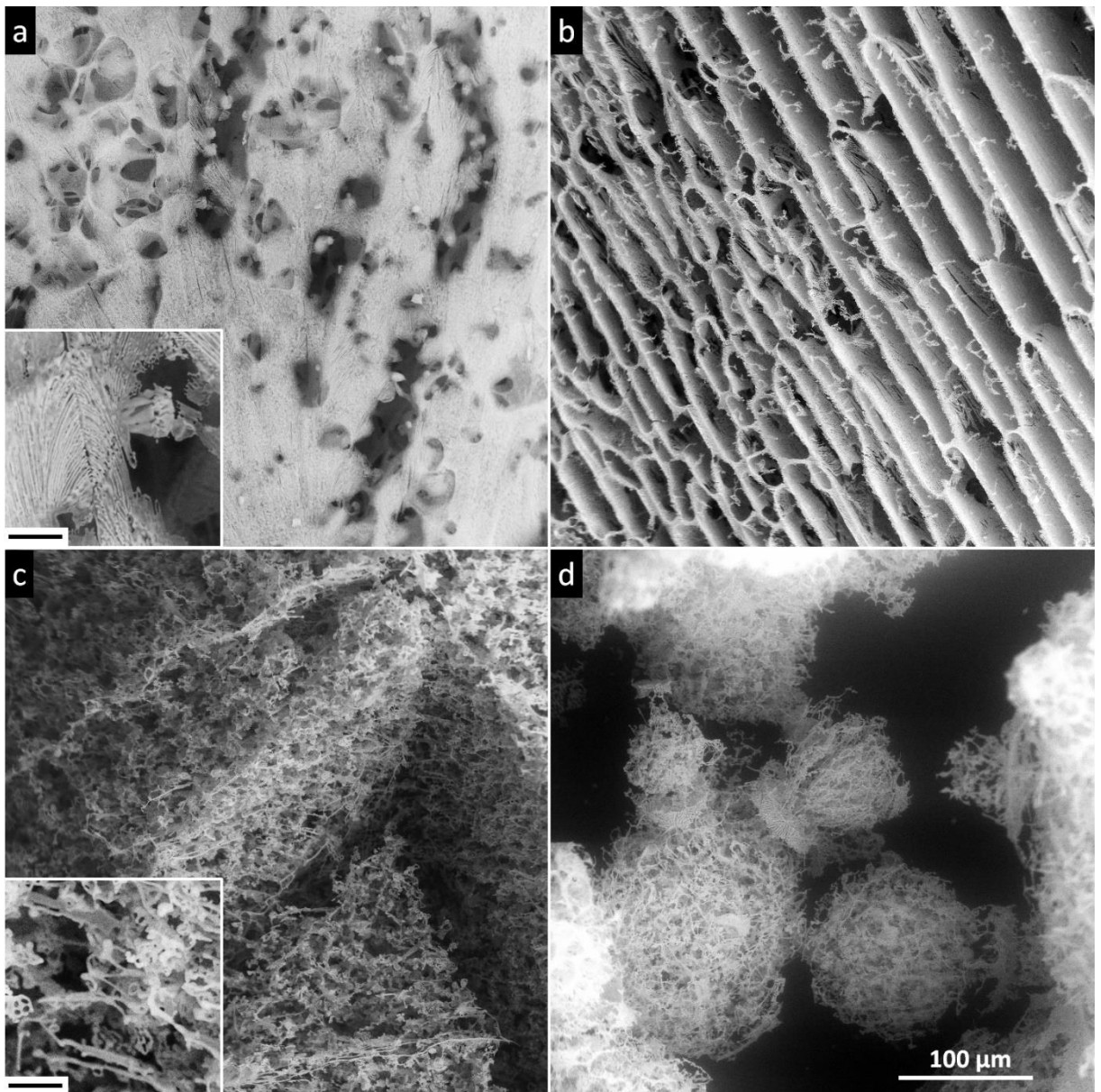

**Figure 13: The salt residua after the sublimation of the frozen samples containing 0.5 M CsCl: (a) a non-seeded droplet, (b) a seeded droplet, (c) an LN-frozen droplet, (d) LN-frozen microspheres. The samples sublimed at −25 °C (below the $T_{eu}$). The scale in panel $d$ applies to all of the images. The corresponding morphologies are detailed in panels $a$ and $c$, where the black bar represents 10 μm.**

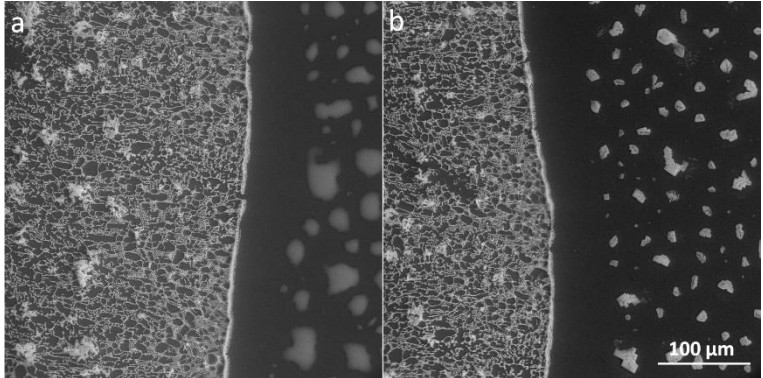

**Figure 14: The supercooled brine beyond the edge of the 0.05 M non-seeded sample (a) and the salt crystals formed via the eventual crystallization of the supercooled brine (b). The original frozen sample had been located in the left side of the panels a and b, its edge is represented by the broader white line (the salt rim).**


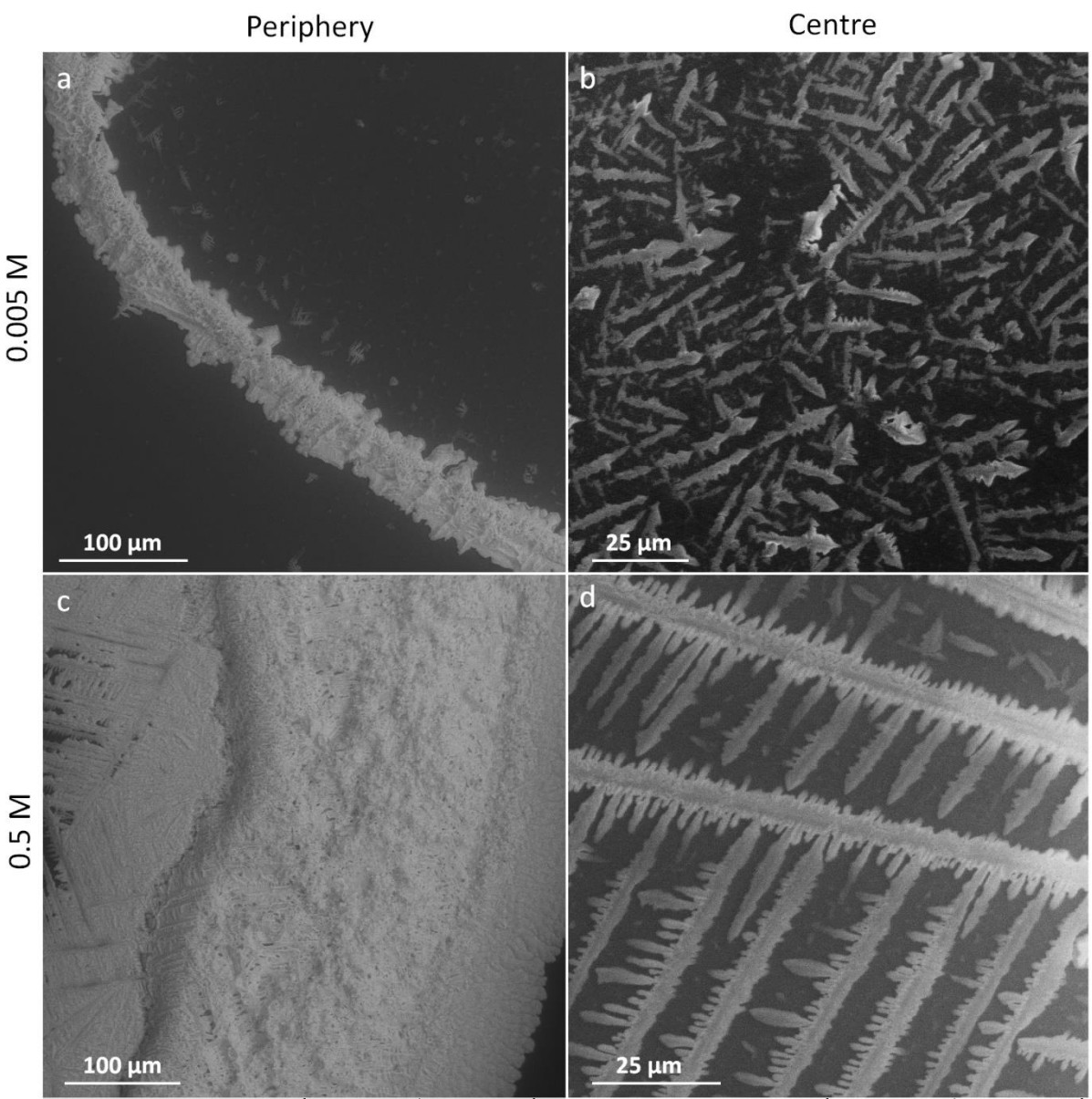

**Figure 15: The structures of the CsCl salt residua after the evaporation of the 0.005 and 0.05 M liquid samples at 2 °C. We imaged spots close to the periphery and the centre of the original sample, respectively.**


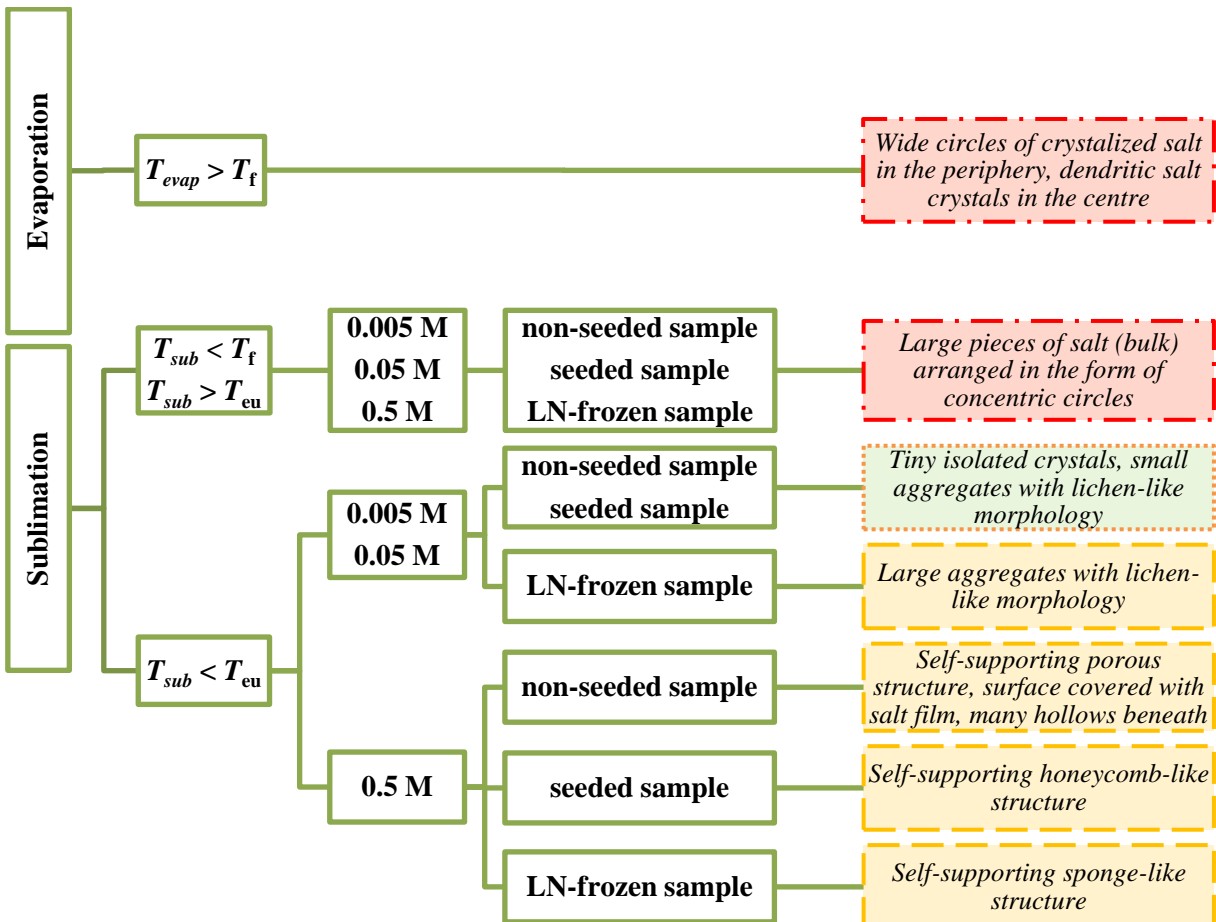

**Figure 16: The structure of the salt residua after the sublimation of the ice, influenced by the sublimation temperature, salt concentration, and freezing method. According to their ability to be windblown and to become aerosols, the salt residua are categorized as follows: readily available (the green box with dotted border), potentially available after their mechanical breakdown (the yellow boxes with dashed borders), improbable (the red box with dash-dotted border). $T_f$, $T_{eu}$, $T_{evap}$, and $T_{sub}$ represent the freezing point, and the eutectic, evaporation and sublimation temperatures, respectively.**

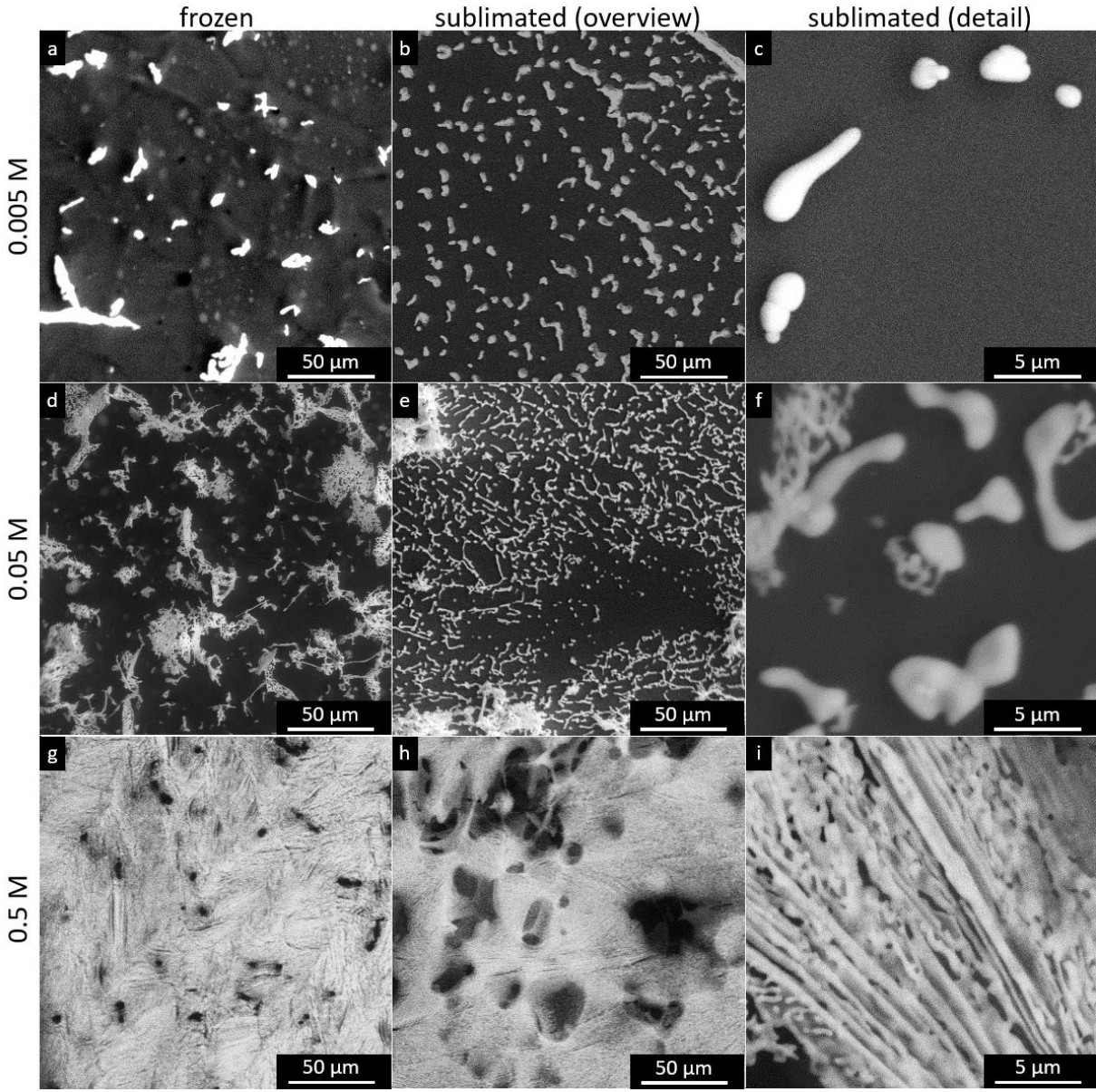

**Figure 17. The structure of the frozen samples (before the sublimation) and the CsCl salt residua after the sublimation of the 0.005, 0.05, and 0.5 M spontaneously frozen samples at −25 °C, i.e. below $T_{eu.}$.**


| | 0.005 M CsCl, non-seeded | 0.005 M CsCl, seeded | 0.05 M CsCl, non-seeded | 0.05 M CsCl, seeded |
|---|---|---|---|---|

| | | | | |
|---|---|---|---|---|
| Surface number density of salt particles / mm⁻² | $3405 \pm 3033$ | $12758 \pm 7017$ | $39200 \pm 25032$ | $59625 \pm 11885$ |
| Volume number density of salt particles / mm⁻³ | $6300 \pm 5613$ | $23604 \pm 12983$ | $72528 \pm 46314$ | $110319 \pm 21989$ |

**Table 1. The calculated surface and volume number densities (mm⁻³) with sample standard deviations under different conditions. The method of the calculation is described in SI (Text S3, Table S1).**

| Concentration | Freezing method | The occurrence of residual salt particles according to their sizes | | | | |
|---|---|---|---|---|---|---|
| | | 1 to 10 μm | 10 to 100 μm | 100 to 200 μm | 200 to 400 μm | > 400 μm |
| 0.005 M | Non-seeded | *** | *** | | | |
| | Seeded | *** | ** | | | |
| | LN-frozen | | | ** | * | *** |
| 0.05 M | Non-seeded | *** | *** | ** | * | |
| | Seeded | *** | *** | *** | *** | |
| | LN-frozen | | | * | ** | *** |
| 0.5 M | Non-seeded | | | | | *** |
| | Seeded | | | | | *** |
| | LN-frozen | | | | | *** |


**Table 2: The distribution of the sizes of the residual salt particles in the samples sublimed at −25 °C. The occurrence of the particles according to their sizes is expressed as follows: *** often; ** occasionally; * seldom.**