# Peer review of "Technical note: Sublimation of frozen CsCl solutions in ESEM: Determining the number and size of salt particles relevant to sea-salt aerosols"

_Atmospheric Chemistry and Physics, 2022_

## Author Comment (AC1)

**Review of "Sublimation of frozen CsCl solutions in ESEM: determining the number and size of salt particles relevant to sea-salt aerosols" submitted to ACP by Vetráková et al.**

**Reviewer #1**

This study presents sublimation of CsCl solution to elucidate formation of fine salt particles and potential as origins of sea-salt aerosols in polar areas. This unique approach can provide us very interesting and important suggestion about sea-salt aerosols in polar areas. There are several interesting conclusions obtained from this study based on laboratory experiments using ESEM:

1) Fine salt particles formed through sublimation of salt solution under colder conditions.

2) Size and number concentrations of salt particles related to temperature and the concentrations of salt solution.

Overall, this topic and subject is suitable for scope of ACP. I am confident that this investigation has potential to demonstrate these interesting results. However, I found some weakness in the manuscript. Hence, I recommend major revision before publication in ACP. My specified suggestions and comments are listed below.

Thank you very much, we really appreciate you consider our study interesting. In the following text, we reply to your comments and answer the questions. We are currently working on implementing the suggested changes into the manuscript.

Major comments

1.  Application of laboratory results to ambient conditions

In this study, sublimation processes were examined under several artificial conditions such as non-seeding, seeding, and LN frozen. Although I understand different conditions between laboratory experiments and fields, more careful discussion are required for robust conclusion. Specifically, authors need to discuss carefully whether the examined artificial conditions are available in polar areas, and what the most usual conditions are in polar areas. For example, there are a large amount of brine on younger sea ice. Actually, the wet surface is present on sea ice even below -20 °C (e.g., Hara et al., ACP, 2017). Therefore, "the edge" might not be realistic in such conditions. On the other hand, salinity of snow on the aged sea ice and multi-year sea ice are lower than that on the younger sea ice, as discussed in this study. However, I am not sure whether "brine" is present in the snow the aged sea ice and multi-year sea ice, or not. Is that available? At least, the snow on the aged sea ice and multi-year sea ice were not wet, as far as I observed that visibly.

Complete freezing of brine is a matter of brine composition and temperature. In our experiments, the eutectic temperature of CsCl brine lies close to −23 °C; below this temperature, brine solidifies (except for the effect in chapter 3.3). However, natural seawater contains versatile ratio of diverse salts and its eutectic temperature depends on particular way of freezing: -54 °C for Ringer-Nelson-Thompson pathway or -36 °C for Gitterman pathway (Gitterman 1937, Vancoppenolle et al. 2019). Therefore, the limit of −25 °C presented in this study (and valid for CsCl) will not be sufficient for the sea salt. However, the temperature of -36 °C, which may be sufficient for solidification of the sea salt brine according to the Gitterman pathway, is often reached in polar areas.

Our experiment result is likely more appliable to the newly formed sea ice surface or with snow on top of it where high salinity is observed. However, on aged sea ice and multi-year sea ice, snow salinity is low and salts are mainly from transported saline particles generated from nearby open leads, polynyas or even open ocean, rather than from bottom interface of the sea ice via upward wicking process (Domine et al., 2004). Therefore, salts are mostly dry not wet in multi-year ice snow, which means the formation process of salt particles from low salinity snow particles may be different from what we observed in this experiment.

The low salinity experimental result may apply to metamorphosed ices; the change of temperatures via a short period of warming or even through the absorption of solar energy due to the existence of impurities such as black carbons and dusts may melt snow (Warren 1984; Kang et al., 2020) and cause a wet brine pond.

Another natural condition under which the sea water of various concentration is freezing embodies in the refreezing of partly melted sea ice. The "wetness" of the snow is given by the salt concentration, its distribution in the medium and the temperature.

2. Seeding

When solid materials are present in brine (i.e., seeding in this study), salts can be crystalized or precipitated on the solid materials. In this study, small ice crystals were used as "seeding" materials. Considering brine is attached to ice under ambient conditions on sea ice and in snow, "seeding" conditions are real situation. Or not? Which are realistic conditions (non-seeding or seeding) which authors considered. Additionally, upside surface of brine on sea ice and snow is attached usually to colder air in polar areas in contrast to colder surface in the bottom (i.e., cool stage) during the laboratory experiments. More careful explanation and discussion are required.

Certainly, heterogenous nucleation by solid particle can trigger the crystallization of a supercooled or (over)saturated solution. In here, we apply ice crystals to trigger the ice crystallization. In the natural arctic conditions brine can appear on the sea ice and snow after its partial melting or flooding of the surface. Then, depending on the climatic conditions such as temperature drop and amount of precipitation, both the spontaneous and seeded freezing can occur. We propose the "seeding" condition is more common than the "non-seeding" condition in the Arctic even though we are not aware of relevant literature. There are numerous items in the polar winter atmosphere that can materialize the seeds, e.g., ice crystals (diamond dusts), drifting or blowing snow particles as well as various natural particles (e.g. dust) and pollutants (e.g. black carbons).

In the polar areas, surface snow or brine exposed to the air on sea ice usually face much colder temperatures than the bottom parts. In the laboratory experiments, only the bottom of the sample is cooled directly by the stage during the freezing phase (the specimen chamber is open to atmospheric conditions). However, the frozen samples are much smaller in comparison with natural ones, therefore we expect only small temperature differences across the sample. In our seeding procedure, the samples are thermally equilibrated prior to seeding, and the seeding crystal is put onto the surface, so we believe the sample freezes from the top, similar to the natural sea ice. On the other hand, the non-seeded samples in our experiments are more likely to freeze from the colder bottom.

When the specimen chamber is closed and evacuated, the gas inside the chamber becomes colder due to its expansion, and the sample surface is cooled via sublimation, too. In this phase of the experiment, the

sample surface may become colder than the bottom part. Still, due to the small sample size, we don't expect large temperature gradients.

3. Evaporation and sublimation

Laboratory experiments showed clearly that many salt particles are observed on the substrates after sublimation and evaporation. This result is very interesting. Crystallization of the salt particles are found on the dry surface on the substrates where brine was absent via brine shift and sublimation as shown in Figs. in the manuscript. The dry surface might be not available under the conditions with presence of brine on sea ice and snow in polar areas.

We agree with the reviewer: there are hardly any dry surfaces in the Arctics. Nevertheless, the observation of the salt particles formation in the microscope is relevant to the natural conditions as the produced particles would fall down to the underlying snow or ice or will be airborne within the sublimation process.

1. Size and size distribution

In this study, maximum diameter is used as particle size. As shown in ESEM images, the morphology of salt particles was irregular not spherical. I understand the difficulty to measure the size of salt particles with irregular shape. However, more careful procedure to measure the size is required, for example, to measure minimum diameter and mean diameter. Also, variations of size distribution of maximum diameter, minimum diameter, and mean diameter should be discussed.

Thank you for your comment. We agree the maximum diameter of particles of irregular shape is not an ideal parameter. We have modified the histograms in Figure 12 and used an equivalent diameter instead. The equivalent diameter expresses the diameter of the disk whose area is equal to the area of the particle. Also, we newly added more parameters related to particle sizes (equivalent, maximum, minimum, and mean diameter) into a statistical summary presented in table S2 in SI.

Specific comments

1. Line 187-189: Add information of Fig. No which you show.

Although we usually show already sublimated samples in the results, during the experiments we observe whole sublimation of frozen solutions and we are able to recognize if the brine is in liquid state or it solidified. In the presented figures, the solidified salt on the ice surface is nicely visible in Figures 11 and 17a,d,g.

Following text was added:

*(the solidified salt on the frozen surface is nicely visible e.g. in Figures 11 and 17a,d,g).*

2. Line 226-227: Size distribution of salt particles by non-seeding and seeding

"Seeding" preferred to form smaller (aerosolable) salt particles in this study. What are reasons of this different? More discussion is useful for readers.

The reasons for this behaviour can be explained by examining the micrographs of the solutions frozen without seeding and with seeding (Figure 2 in Vetráková 2019) and micrographs of the appropriate sublimation residua at both discussed concentrations (Figure 7 and 8 in the current manuscript). The explanation lies in the freezing directionality and rate. With the seeding, the sample freezes slowly at low degree of supercooling forming larger ice crystals (Table 2 in Vetráková 2019). In contrast, non-seeded samples freeze faster (ca 150 mm/s) and directionally from the bottom up; such a process results in larger amount of salt on the surface. Micrographs show clearly (Figure 7 and 8), that in seeded samples the salt remained mostly in the veins finally forming regular geometrical patterns after the ice sublimation, whereas the non-seeded samples allowed mostly the larger particles originating in the triple junctions, pools and on the ice surface.

Section 3.2: LN-frozen and full sublimation

LN-frozen samples were simulated with the complete freezing condition. Difference between LN-frozen samples and the other samples is very interesting to understand formation of small salt particles depending on freezing processes under the colder conditions. However, I wonder whether the complete freezing of brine can proceed in polar areas, particularly where sea-salt aerosols are released. As well as, I am not sure that full sublimation occur on the surface under the conditions with presence of brine. Add more careful explanation and discussion.

In the manuscript, the LN-freezing experiments are present mostly to show the large difference between "natural" conditions (spontaneous freezing and seeding) and laboratory freezing in LN. Moreover, LN temperatures are relevant for extraterrestrial bodies. Complete freezing of brine is a matter of brine composition and temperature. In our experiments, the eutectic temperature of CsCl brine lies close to −23 °C; below this temperature, brine solidifies (except for the effect in chapter 3.3). However, natural seawater contains versatile ratio of diverse salts and its eutectic temperature depends on particular way of freezing: -54 °C for Ringer-Nelson-Thompson pathway or -36 °C for Gitterman pathway (Gitterman 1937, Vancoppenolle et al. 2019). The temperature of -36 °C is often reached in polar areas. In our experiments presented in Zavacka (2022) we observed the differences between the sublimation at -30 and -40 °C indicative of remaining liquid at higher temperature.

At low temperature (solidified brine) full sublimation is not needed for the release of salt particles. Small salt particles are released continually as the ice sublimates.

1. Section 3.5: Surface conditions of sea ice with brine and snow

In this section, implication to polar atmosphere and conditions was mentioned and discussed. Surface conditions in sea ice areas changed drastically depending on presence of brine, sea ice growth (aging), accumulation of snowfall, and so on, as shown by Hara et al. (2017). Similar to sea ice conditions, snow salinity (i.e., sea-salt concentrations) on sea-ice is varied largely by blowing snow and snowfall. Therefore, I strongly recommend that formation of aerosolable salt particles is discussed on every (or typical) surface conditions such as wet surface on younger sea ice by brine, conditions with presence of frost flower, wet saline snow on sea-ice, and low – middle saline snow on sea ice (aged seasonal sea ice or multi – year sea ice). Although some explanation and discussion were mentioned already in the manuscript, correspondence between laboratory results and the ambient conditions in the polar areas was interpreted hardly because of partly confusion.

The frost flowers and young sea ice with salt concentration above 35 psu are unlikely direct sources of SSA at any temperature. However, dilution of the salt and the presence of the ice and snow with concentration below 8 psu (as is typical of saline snow on sea ice) make the formation of SSA feasible especially at low temperature.

Line 367: Temperature of mirabilite precipitation

Add reference which show the temperature of -6.4 °C.

The reference was added to the text.

*When the ice temperature ranges below −6.4°C, which is a typical situation in a polar winter and spring, mirabilite ($Na_2SO_4 \cdot 10\ H_2O$) starts to precipitate out from the brine (Butler et al., 2016a, 2016b), …*

Line 369-373:

Changes of surface conditions of sea-ice were shown in Hara et al. (2017).

Hara, K., Matoba, S., Hirabayashi, M., and Yamasaki, T.: Frost flowers and sea-salt aerosols over seasonal sea-ice areas in northwestern Greenland during winter–spring, Atmos. Chem. Phys., 17, 8577-8598, https://doi.org/10.5194/acp-17-8577-2017, 2017

Thank you for the suggestion, we added the reference to the text.

Sizes of small salt particles shown in this study ranged in micrometer or larger than 10 mm. It is true that particles with size of micrometers are "aerosolable". However, these particles may be attached to the surface of ice and snow in polar regions. What are specific processes to release sea-salt aerosols into the atmosphere? Because of less volatility of sea-salts, sea-salt aerosols must be released through the physical processes such as erosion during the strong winds and then sublimation. Although release of salt particles with size of micrometers can be explained by the processes, sea-salt aerosols in the Antarctic were distributed in smaller than 1 mm (Yang et al., 2019; Frey et al., 2020) and smaller than 100 nm (Hara et al., 2011). If possible, these issues should be discussed.

We performed most our experiments using the resolution of 0.5 µm, which enabled us to get a full picture of the related processes. When more detailed pictures were recorded, we also observed smaller particles, but we are not able to evaluate their number and size due to lack of statistically relevant data.
* * *
References:

Domine, F., Sparapani, R., Ianniello, A., and Beine, H. J.: The origin of sea salt in snow on Arctic sea ice and in coastal regions, Atmos. Chem. Phys., 4, 2259–2271, https://doi.org/10.5194/acp-4-2259-2004, 2004.

Gitterman, K. (1937). Thermitcheskii analismorskoi vody (Thermal analysis of sea water) (*CRREL-TL-287*). Hanover, NH: Trudy Solyanoy. Laboratorii (SSSR), translation from 1971, USA Cold Regions Research and Engineering Laboratory

Kang, S., Zhang, Y., Qian, Y. & Wang, H. A review of black carbon in snow and ice and its impact on the cryosphere. *Earth-Science Rev.* **210**, 103346 (2020).

Vancoppenolle, M.; Madec, G.; Thomas, M.; McDougall, T. J., Thermodynamics of Sea Ice Phase Composition Revisited. *Journal of Geophysical Research: Oceans* **2019,** *124* (1), 615-634.

Warren, S. (1984). Impurities in Snow: Effects on Albedo and Snowmelt (Review). Annals of Glaciology, 5, 177-179. doi:10.3189/1984AoG5-1-177-179

Závacká, K. *et al.* Temperature and Concentration Affect Particle Size Upon Sublimation of Saline Ice: Implications for Sea Salt Aerosol Production in Polar Regions. *Geophys. Res. Lett.* **49**, 1–10 (2022).

---

## Author Comment (AC2)

**Review of "Sublimation of frozen CsCl solutions in ESEM: determining the number and size of salt particles relevant to sea-salt aerosols" submitted to ACP by Vetráková et al.**

**Reviewer #2**

**General comments:**

Vetrákova et al. present a thorough study of sublimation from CsCl/water in an environmental SEM (ESEM). The research is well motivated, carried out with expertise, and features a good discussion. It is of good value for the scientific community, including atmospheric science. It is also well written, although slightly too much targetted to specialists.

The paper is rather large, and features a plethora of images and topics, and the results are described in great detail. It could profit either from substantial cutting, putting a narrower focus, or - in the other extreme - a more complete view. This would consist the "revision", while scientifically almost everything is very sound (see below).

In case of a more complete view, valuable information would be the phase diagrams NaCl/ice/water and CsCl/ice/water, discussions on crystal morphology of all species (would that include NaCl.2H2O?), a discussion of the regular patterns (e.g. fig. 7c), comparison to simple optical microscopy studies, (semi-)quantitative evaluations, etc.

Thank you very much for your comments. In the following text, we reply to your comments and answer the questions. We are currently working on implementing the suggested changes into the manuscript.

**Specific comments:**

It is not always clear how much the paper builds on results from Závacká et al. (2022), and what is new (except of course the cation, and the advantages of using a heavy ion).

The Letter by Závacká et al. (2022) shows sublimation residua of sea salts for a range of concentrations and sublimation temperatures (−16, −30, −40 °C). We observed low temperature was needed for formation of small salt particles, however, due to the complexity of the sea salt, difficulty in defining eutectic temperature, and its poor visibility on the ice surface, the relation between formation of small particles and eutectic solidification was not straightforward.

All the samples in Závacká et al. (2022) were prepared only by one freezing method (spontaneous freezing). In the present manuscript we wish to detail three ways of sample freezing, compare the results (size and number of particles) in terms of freezing method (rate and directionality of freezing, …).

The observation of crystallization of CsCl below and above the $T_{eu}$ allowed us to present hypotheses about the mechanisms how the particles are formed.

The discussion and interpretation of figure 2 is, compared to all others, very short. Specifically, the droplet-like features on figures 2a and 2b require an explanation. Rather few readers are experts in ESEM, let alone ESEM of multiphase systems!

Thank you for your comment, the features are newly described in more detail.

The brine (section 3.3) must be supercooled, and indeed the droplet-like features suggest the liquid state. But is there any independent proof of the liquid nature?

We infer the liquid state of the droplet-like features from their visual appearance, and from their transformation to salt crystals during the observation. We did not use any further detection methods to proof their liquid state.

**Technical corrections:**

The surface pretreatments are not provided. They are very important for the Peltier surface and for the silicon wafer (for which the reader requires additionally how it was fixed and thermally coupled to the Peltier).

We provided the details of the silicon surface of the Peltier stage. Following text was added to the chapter 2.1.

*Surface of our cooling stage is made from very pure, commercially available silicon wafer that is usually used for the production of semiconductor components. The wafer had no additional surface treatment. It was glued to the Peltier cell with a highly thermally conductive adhesive, compatible with the low temperature and reduced gas pressure environment of the microscope.*

Line 137: 650 Pa should not be called "ambient" pressure, which is ca. 101000 Pa.

The "ambient pressure" refers to the pressure around the sample inside the specimen chamber of the ESEM. The term is used in the meaning of "surrounding". It shall not express the atmospheric pressure. We can use the term "chamber pressure" instead.

Line 186: The temperature sensor and its setup are not described.

Following text was added to chapter 2.1:

*Due to confined space and electrical interference between the temperature sensor and the detector, it is not able to directly measure the temperature on the surface of the sample during the experiment inside the ESEM chamber. Therefore, the sample temperature is inferred from the temperature of the Peltier cooler. The actual sample temperature was validated outside the ESEM at atmospheric conditions using Pt1000 temperature sensor (P1K0.161.6W.A.010, Innovative sensor technology IST AG, Switzerland) frozen inside the sample. The bias between the temperature of the Peltier cooler and the actual sample temperature was no more that 2 °C.*

---

## Author Response (AR1)

We thank the referees for their helpful, constructive comments, which enabled us to reconsider our interpretations, discuss some details more clearly, and eliminate relevant problems. We hope our replies to the comments, answers for the questions, and the modifications of the manuscript will help to clarify the ambiguities to reviewers' satisfaction. Our responses to the comments are outlined in blue, and the modified portions of the manuscript are shown in *italics*.

**Reviewer #1**

This study presents sublimation of CsCl solution to elucidate formation of fine salt particles and potential as origins of sea-salt aerosols in polar areas. This unique approach can provide us very interesting and important suggestion about sea-salt aerosols in polar areas. There are several interesting conclusions obtained from this study based on laboratory experiments using ESEM:

1) Fine salt particles formed through sublimation of salt solution under colder conditions.

2) Size and number concentrations of salt particles related to temperature and the concentrations of salt solution.

Overall, this topic and subject is suitable for scope of ACP. I am confident that this investigation has potential to demonstrate these interesting results. However, I found some weakness in the manuscript. Hence, I recommend major revision before publication in ACP. My specified suggestions and comments are listed below.

Thank you for the valuable comments; we appreciate the fact that you consider our study interesting. In the text below, we propose responses to your assessment and discuss the questions. The suggested changes have been implemented in the manuscript.

**Major comments**

**Application of laboratory results to ambient conditions**

In this study, sublimation processes were examined under several artificial conditions such as non-seeding, seeding, and LN frozen. Although I understand different conditions between laboratory experiments and fields, more careful discussion are required for robust conclusion. Specifically, authors need to discuss carefully whether the examined artificial conditions are available in polar areas, and what the most usual conditions are in polar areas. For example, there are a large amount of brine on younger sea ice. Actually, the wet surface is present on sea ice even below -20 °C (e.g., Hara et al., ACP, 2017). Therefore, "the edge" might not be realistic in such conditions. On the other hand, salinity of snow on the aged sea ice and multi-year sea ice are lower than that on the younger sea ice, as discussed in this study. However, I am not sure whether "brine" is present in the snow the aged sea ice and multi-year sea ice, or not. Is that available? At least, the snow on the aged sea ice and multi-year sea ice were not wet, as far as I observed that visibly.

Thank you! This comment allowed us to realize that the manuscript lacked an appropriate connection to the environmental conditions; we have included this aspect in numerous passages of the text. We have expanded especially the Chapters 3.5 and 3.6 in this manner.

Our experimental results are likely more applicable to the newly formed sea ice surface or with snow on top of it where high salinity is observed. However, on aged sea ice and multi-year sea ice, snow salinity is low and salts are mainly from transported saline particles generated from nearby open leads, polynyas or even open ocean, rather than from bottom interface of the sea ice via upward wicking process (Domine et al., 2004). Therefore, salts are mostly dry not wet in multi-year ice snow, which correspond to low-temperatures observed in our experiments.

The low salinity experimental result may apply to metamorphosed ices; the change of temperatures via a short period of warming or even through the absorption of solar energy due to the existence of impurities such as black carbons and dusts may melt snow (Warren 1984; Kang et al., 2020) and cause a wet brine pond. That may be the mechanism of brine formation in the snow.

Another natural condition under which the sea water of various concentration is freezing embodies in the refreezing of partly melted sea ice.

The "wetness" of the snow, or the amount of the available water in the ice, is not easily estimated by microscopic means, rather it is given by the concentration of all dissolved components (inorganic and organic), its distribution in the medium, temperature, and ice crystal sizes. It seems that the phase diagram is not sufficient to describe the solution phase in ice as the effects of small veins dimensions may decrease the temperature further as described by Gibbs-Thomson equation (Thangswamy et al., 2018). However, for the first approximation, we can consider just the phase behaviour. In our experiments, the eutectic temperature of CsCl brine lies close to −23 °C; below this temperature, brine solidifies (except for the effect in chapter 3.3). However, natural seawater contains versatile ratio of diverse salts and its eutectic temperature depends on particular way of freezing: -54 °C for Ringer-Nelson-Thompson pathway or -36 °C for Gitterman pathway (Gitterman 1937, Vancoppenolle et al. 2019). Thus, the sublimation results of frozen CsCl solutions performed at −25 °C inhere are comparable to those of frozen seawater at −40 °C (Závacká et al., 2022).The temperature of −36 °C, which shall be sufficient for solidification of the brine according to this pathway, is often reached in polar areas.

*The denoted molar concentrations of 0.005, 0.05, and 0.5 M CsCl are equivalent to the molar concentrations of 0.29, 2.9, and 28 psu NaCl solutions, respectively. The NaCl psu equivalents are listed for a straightaway comparison to the seawater salinity. These concentration values were applied to mimic the broad range of salinities in the environment…*

*We showed that the formation of the small particles is restricted or very limited if the brine is liquid during the ice sublimation, i.e., at temperatures higher than the $T_{eu}$: that are the conditions typical of the young sea ice with or without the frost flowers.*

*LN-frozen samples behaved differently, but these conditions are not likely to represent those in the polar areas; they are relevant for extraterrestrial bodies (Fox-Powell and Cousins, 2021).*

*However, it should be borne in mind that the temperature relative to $T_{eu}$ must be considered in the comparisons; at −25 °C CsCl solution is below its eutectic point, whereas the eutectic point of sea water is either at −36 °C or at −54 °C (Vancoppenolle et al., 2019).*

*The temperature of −36 °C, which ought to suffice for solidifying the brine according to this pathway, is often reached in polar winter-spring. Moreover, below −23 °C a fraction of solidified salt abruptly increases*

(11% at −23; 58% at −25; 85% at −33; 100% at −36 °C (Vancoppenolle et al., 2019)). We can speculate that, under the conditions of sequential crystallisation of sea salts, full solidification may not be needed to release of a low amount of SSA. Spatial separation of the crystallised salts and the remaining (low amount of) liquid brine, i.e., segregation by drainage of the brine to lower stages, might be sufficient for releasing some aerosolizable particles from the surface of the ice or snow.

**Seeding**

When solid materials are present in brine (i.e., seeding in this study), salts can be crystalized or precipitated on the solid materials. In this study, small ice crystals were used as "seeding" materials. Considering brine is attached to ice under ambient conditions on sea ice and in snow, "seeding" conditions are real situation. Or not? Which are realistic conditions (non-seeding or seeding) which authors considered. Additionally, upside surface of brine on sea ice and snow is attached usually to colder air in polar areas in contrast to colder surface in the bottom (i.e., cool stage) during the laboratory experiments. More careful explanation and discussion are required.

Certainly, heterogenous nucleation by solid particle can trigger the crystallization of a supercooled or (over)saturated solution. In here, we apply ice crystals to trigger the ice crystallization. In the natural arctic conditions brine can appear on the sea ice and snow after its partial melting or flooding of the surface. Then, depending on the climatic conditions such as air temperature and amount of ice nuclei particle (INP), both the spontaneous and seeded freezing can occur.

In polar regions, surface snow or brine exposed to the air on sea ice usually face much colder temperatures than the bottom parts, which is different to the laboratory experiment conditions, where the bottom of the sample is cooled directly by the stage during the freezing phase. However, the frozen samples are much smaller in comparison with natural ones, therefore we expect only small temperature differences across the sample. In addition, in our seeding procedure, the samples are thermally equilibrated prior to seeding, and the seeding crystal is put onto the surface, so we believe the sample freezes from the top. On the other hand, the non-seeded samples are more likely to freeze from the colder bottom. When the specimen chamber is closed and evacuated, the gas inside the chamber becomes colder due to its expansion, and the sample surface is cooled via sublimation, too. In this phase of the experiment, the sample surface may become colder than the bottom part. Still, due to the small sample size, we don't expect large temperature gradients.

Then, depending on the climatic conditions, including the air temperature and amount of aerosols, both the spontaneous and seeded freezing can occur. We propose that the "seeding" condition is more common than the "spontaneous, or non-seeding" condition in the Arctic. There are numerous items in the polar winter atmosphere that can materialize the seeds, e.g., ice crystals (diamond dusts), drifting or blowing snow particles, and various natural and man-made ice nucleating particles (e.g., dust, pollutants, black carbons). The artificial and natural freezing of seeded samples will differ in the temperature gradients: In the polar areas, surface snow or brine exposed to the air on sea ice usually face much colder temperatures than the bottom parts; thus, a large temperature gradient would be experienced across the sample. On the other hand, the temperature of the artificial samples is much more uniform, as the samples are thermally equilibrated prior to seeding and their overall size is very small.

**Evaporation and sublimation**

Laboratory experiments showed clearly that many salt particles are observed on the substrates after sublimation and evaporation. This result is very interesting. Crystallization of the salt particles are found

on the dry surface on the substrates where brine was absent via brine shift and sublimation as shown in Figs. in the manuscript. The dry surface might be not available under the conditions with presence of brine on sea ice and snow in polar areas.

A dry surface might be available at low temperatures when almost all the brine had solidified, or after brine drainage through pores in the ice. In the field observations, a dry surface was exemplified by very old sea ice covered with snow (Hara et al., 2017).

**Size and size distribution**

In this study, maximum diameter is used as particle size. As shown in ESEM images, the morphology of salt particles was irregular not spherical. I understand the difficulty to measure the size of salt particles with irregular shape. However, more careful procedure to measure the size is required, for example, to measure minimum diameter and mean diameter. Also, variations of size distribution of maximum diameter, minimum diameter, and mean diameter should be discussed.

Thank you for your comment. We agree the maximum diameter of particles of irregular shape is not an ideal parameter. We have modified the histograms in Figure 12 and used an equivalent diameter instead. The equivalent diameter expresses the diameter of the disk whose area is equal to the area of the particle. Also, we newly added more parameters related to particle sizes (equivalent, maximum, minimum, and mean diameter) into a statistical summary presented in Table S2 in SI.

*Equivalent diameter was chosen to express the size of the residual particles; it represents a diameter of a disk whose area is equal to the area of the particle. The majority of the particles were less than 10 μm in the equivalent diameter (Figure 12).*

**Specific comments**

**Line 187-189: Add information of Fig. No which you show.**

Although we usually show already sublimated samples in the results, during the experiments we observe whole sublimation of frozen solutions and we are able to recognize if the brine is in liquid state or it solidified.  In the presented figures, the solidified salt on the ice surface is nicely visible in Figures 11 and 17a,d,g.

The following text has been added:

*(the solidified salt on the frozen surface is nicely visible e.g. in Figures 11 and 17a,d,g).*

**Line 226-227: Size distribution of salt particles by non-seeding and seeding**

"Seeding" preferred to form smaller (aerosolable) salt particles in this study. What are reasons of this different? More discussion is useful for readers.

The reasons for this behaviour can be explained by examining the micrographs of the solutions frozen without and with seeding (Figure 2 in Vetráková 2019) and through exploring the micrographs of the appropriate sublimation residua at both of the discussed concentrations (Figure 7 and 8 in the current manuscript). The explanation lies in the freezing directionality and rate. With the seeding, the sample

freezes slowly at a low degree of supercooling, forming larger ice crystals (Table 2 in Vetráková 2019). In contrast, the non-seeded samples freeze faster (ca 150 mm/s) and directionally from the bottom up; such a process results in a larger amount of salt on the surface. The micrographs show clearly (Figure 7 and 8) that in the seeded samples the salt remained mostly in the veins, finally forming regular geometrical patterns after the ice sublimation, whereas the non-seeded samples produced mostly the larger particles in the triple junctions, pools, and on the ice surface.

The following text has been added to the manuscript:

*Moreover, the seeded samples generally produced smaller salt particles than the non-seeded samples (Figure 12, Table S2). The explanation of this behaviour may lie in the freezing directionality and rate. With the seeding, the sample freezes slowly at low degree of supercooling forming larger ice crystals* (Vetráková et al., 2019)*. In contrast, non-seeded samples freeze faster (ca 150 mm s$^{-1}$) and directionally from the bottom up; such a process results in larger amount of salt on the surface. Micrographs in the figures 7 and 8 clearly show the residua in the seeded samples forming regular geometrical patterns, indicating the salt remained mostly in the veins after freezing, whereas the non-seeded samples allowed mostly the larger particles originating in the triple junctions, pools and on the ice surface. Regarding larger number of the particles and their smaller size, seeding favours the formation of aerosolable salt particles significantly more than spontaneous freezing.*

**Section 3.2: LN-frozen and full sublimation**

LN-frozen samples were simulated with the complete freezing condition. Difference between LN-frozen samples and the other samples is very interesting to understand formation of small salt particles depending on freezing processes under the colder conditions. However, I wonder whether the complete freezing of brine can proceed in polar areas, particularly where sea-salt aerosols are released. As well as, I am not sure that full sublimation occur on the surface under the conditions with presence of brine. Add more careful explanation and discussion.

In the manuscript, the LN-freezing experiments are present mostly to expose the large difference between the "natural" conditions (spontaneous freezing and seeding) and laboratory freezing in LN. Moreover, LN temperatures are relevant for extraterrestrial bodies. Complete freezing of the brine is a matter of the brine composition and temperature. In our experiments, the eutectic temperature of CsCl brine lies close to −23 °C; below this limit, the brine solidifies (except for the effect in Chapter 3.3). However, natural seawater contains diverse salts, and its eutectic temperature depends on the particular way of freezing: -54 °C for the Ringer-Nelson-Thompson pathway or -36 °C for the Gitterman pathway (Gitterman 1937, Vancoppenolle et al. 2019). The temperature of -36 °C is often reached in polar areas. In our experiments presented in Zavacka (2022), we observed differences between the sublimation at -30 and -40 °C, this effect being indicative of liquid remaining at the higher temperature.

At the lower temperature (solidified brine), full sublimation is not needed to release salt particles. Small salt particles are released continually as the ice sublimes.

The following texts were added to chapter 3.5:

*However, natural seawater contains diverse salts, and its eutectic temperature depends on the particular way of freezing: −54 °C for the Ringer-Nelson-Thompson pathway or −36 °C for the Gitterman pathway*

*(Marion et al., 1999; Vancoppenolle et al., 2019). Recent experimental and modelling data support the latter as the reference equilibrium pathway (Vancoppenolle et al., 2019 and the references herein); thus, the sublimation results of the frozen CsCl solutions acquired at −25 °C are comparable to those of frozen seawater at −40 °C (Závacká et al., 2022). The temperature of −36 °C, which ought to suffice for solidifying the brine according to this pathway, is often reached in polar winter-spring. Moreover, below −23 °C a fraction of solidified salt abruptly increases (11% at −23; 58% at −25; 85% at −33; 100% at −36 °C (Vancoppenolle et al., 2019)). We can speculate that, under the conditions of sequential crystallisation of sea salts, full solidification may not be needed to release of a low amount of SSA. Spatial separation of the crystallised salts and the remaining (low amount of) liquid brine, i.e., segregation by drainage of the brine to lower stages, might be sufficient for releasing some aerosolizable particles from the surface of the ice or snow.*

*We related the absence of liquid brine in the ice to the formation of the fine particles; these were released continually, as the ice sublimed at sufficiently low temperature.*

*LN-freezing conditions are not likely to represent those in the polar areas; they are relevant for extra-terrestrial bodies.*

**Section 3.5: Surface conditions of sea ice with brine and snow**

In this section, implication to polar atmosphere and conditions was mentioned and discussed. Surface conditions in sea ice areas changed drastically depending on presence of brine, sea ice growth (aging), accumulation of snowfall, and so on, as shown by Hara et al. (2017). Similar to sea ice conditions, snow salinity (i.e., sea-salt concentrations) on sea-ice is varied largely by blowing snow and snowfall. Therefore, I strongly recommend that formation of aerosolable salt particles is discussed on every (or typical) surface conditions such as wet surface on younger sea ice by brine, conditions with presence of frost flower, wet saline snow on sea-ice, and low – middle saline snow on sea ice (aged seasonal sea ice or multi – year sea ice). Although some explanation and discussion were mentioned already in the manuscript, correspondence between laboratory results and the ambient conditions in the polar areas was interpreted hardly because of partly confusion.

We have extended the discussion on formation of aerosolizable salt particles in relation with typical surfaces in polar areas. The followings text were added to the chapter 3.6:

*Thus, **the frost flowers are not assumed to be an effective source of SSAs**, due to their higher temperature and very high salinity. Highly saline **young sea ice** is not a probable source of SSAs for similar reasons; however, the release of ikaite-like and mirabilite-like particles was detected also from fresh sea ice areas, which are supposed to be wet-surfaced, even though the details of the process remain unclear (Hara et al., 2017).*

*Normally, young sea ice is thin and thus relatively warmer than thick ice, and a temperature as low as the $T_{eu}$ could be difficult to reach on very thin young ice. Therefore, **the snowpack on multi-year ice is more likely to form numerous fine salt structures and SSAs** than the snow on relatively thin sea ice. This study favours (almost) **dry, low-salinity surfaces** when concerning the release of aerosolizable salt particles. Such a scenario is in accordance with the field observations, where all the low-salinity snow eroded from the dry surface of old sea ice by strong winds, whereas a large amount of the snow remained on young sea ice because of wet conditions (Hara et al., 2017). Thus, multi-year sea ice and a low salinity snow lying*

*on the sea ice (with typical salinities below 3 psu) make the formation of SSAs feasible especially at low temperatures, when the brine is already crystalline and the wetness of the surface remains very small.*

*If the **salt concentration is very low**, e.g., below 0.085 psu (Závacká et al., 2022), as is the case of snow sputtered by sea-salts (Dominé et al., 2003) and snow on one-year sea ice (in the Antarctic, ~40% of such snow has a salinity of < 0.1 psu (Massom et al., 2001)), the formation of aerosolizable particles by strong wind agitation is not excluded even at temperatures above the $T_{eu}$, because the brine is scattered throughout the snow matrix; such snow with a very low amount of liquid brine may be airborne, and its full sublimation allows the formation of fine salt particles, as there is not enough salt to coalesce to larger pieces.*

**Line 367: Temperature of mirabilite precipitation**

Add reference which show the temperature of -6.4 °C.

The following references showing the mirabilite crystallization temperature were added:

Butler, B. M., Papadimitriou, S., Santoro, A. and Kennedy, H.: Mirabilite solubility in equilibrium sea ice brines, Geochim. Cosmochim. Acta, 182, 40–54, doi:10.1016/j.gca.2016.03.008, 2016a.

Butler, B. M., Papadimitriou, S. and Kennedy, H.: The effect of mirabilite precipitation on the absolute and practical salinities of sea ice brines, Mar. Chem., 184, 21–31, doi:10.1016/j.marchem.2016.06.003, 2016b.

*When the ice temperature ranges below −6.4°C mirabilite ($Na_2SO_4 \cdot 10\ H_2O$) starts to precipitate out from the brine (Butler et al., 2016a, 2016b),…*

**Line 369-373: Changes of surface conditions of sea-ice were shown in Hara et al. (2017).**

Hara, K., Matoba, S., Hirabayashi, M., and Yamasaki, T.: Frost flowers and sea-salt aerosols over seasonal sea-ice areas in northwestern Greenland during winter–spring, Atmos. Chem. Phys., 17, 8577-8598, https://doi.org/10.5194/acp-17-8577-2017, 2017

Thank you for the suggestion, we have added the references to the publication to numerous places in the text:

*Highly saline **young sea ice** is not a probable source of SSAs for similar reasons; however, the release of ikaite-like and mirabilite-like particles was detected also from fresh sea ice areas, which are supposed to be wet-surfaced, even though the details of the process remain unclear (Hara et al., 2017).*

*The stages of frost flowers' formation, growth, and erosion by winds were described by Hara et al., suggesting a release of Mg-rich salt particles from the surface snow that had covered the aged frost flowers (Hara et al., 2017).*

*Such a scenario is in accordance with the field observations, where all the low-salinity snow eroded from the dry surface of old sea ice by strong winds, whereas a large amount of the snow remained on young sea ice because of wet conditions (Hara et al., 2017).*

Sizes of small salt particles shown in this study ranged in micrometer or larger than 10 mm. It is true that particles with size of micrometers are "aerosolable". However, these particles may be attached to the surface of ice and snow in polar regions. What are specific processes to release sea-salt aerosols into the atmosphere? Because of less volatility of sea-salts, sea-salt aerosols must be released through the physical processes such as erosion during the strong winds and then sublimation. Although release of salt particles with size of micrometers can be explained by the processes, sea-salt aerosols in the Antarctic were distributed in smaller than 1 mm (Yang et al., 2019; Frey et al., 2020) and smaller than 100 nm (Hara et al., 2011). If possible, these issues should be discussed.

Thank you for your comment, we newly discuss these issues in the text:

*It is clear that strong winds are required to lift either partially wet saline snow particles at temperature above the $T_{eu}$ from the snow/ice surfaces to the air, where salt particles can be formed through the loss of water vapour via evaporation or sublimation processes, or to directly erode these already crystallized salt particles from snow/ice surface at temperature below the $T_{eu}$. Submicron-sized and ultrafine (<100 nm) SSAs were detected near Antarcica (Frey et al., 2020; Hara et al., 2011) and also in the central Arctic during the recent MOSAiC (The Multidisciplinary drifting Observatory for the Study of Arctic Climate expedition) field campaign. The enhancements of ultrafine SSAs observed in the central Arctic were mostly associated with blowing snow events (Gong et al., Nature Geoscience, in review), which is in good agreement with the model calculation, as the model already predicted ultrafine SSAs (diameter <200 nm) produced by blowing snow (Yang et al., 2019). We performed most of our experiments with the resolution of 0.5 μm, which enabled us to obtain a full picture of the related processes, but the detection of submicron and ultrafine particles was hindered. In higher resolution images we occasionally observed the submicron particles but have been unable to evaluate their number and size due to a lack of statistically relevant data.*

**Reviewer #2**

**General comments:**

Vetrákova et al. present a thorough study of sublimation from CsCl/water in an environmental SEM (ESEM). The research is well motivated, carried out with expertise, and features a good discussion. It is of good value for the scientific community, including atmospheric science. It is also well written, although slightly too much targetted to specialists.

The paper is rather large, and features a plethora of images and topics, and the results are described in great detail. It could profit either from substantial cutting, putting a narrower focus, or - in the other extreme - a more complete view. This would consist the "revision", while scientifically almost everything is very sound (see below).

In case of a more complete view, valuable information would be the phase diagrams NaCl/ice/water and CsCl/ice/water, discussions on crystal morphology of all species (would that include NaCl.2H2O?), a discussion of the regular patterns (e.g. fig. 7c), comparison to simple optical microscopy studies, (semi-)quantitative evaluations, etc.

Thank you for the comments. Below we offer replies to your comments and responses to the questions. The suggested changes have been incorporated into the manuscript, including an expansion of the text with additional explanatory passages.

We have extended the discussion in terms of environmental applications. We hope that the extended discussion now relates the laboratory experiments to the naturally occurring phenomena.

We have constructed the phase diagrams of CsCl-$H_2O$ and NaCl-$H_2O$ systems previously, so we provide the references:

*The phase diagrams of CsCl-$H_2O$ and NaCl-$H_2O$ systems were presented in our previous publications (Vetráková et al., 2019; Yang et al., 2017).*

**Specific comments:**

**It is not always clear how much the paper builds on results from Závacká et al. (2022), and what is new (except of course the cation, and the advantages of using a heavy ion).**

The Letter by Závacká et al. (2022) shows sublimation residua of sea salts at a range of concentrations and sublimation temperatures (−16, −30, −40 °C). We observed that a low temperature was needed to form small salt particles; however, due to the complexity of the sea salt, difficulty in defining the eutectic temperature, and poor visibility of the salt on the ice surface, the relationship between the formation of small particles and eutectic solidification was not definable in a straightforward manner.

All the samples in Závacká et al. (2022) were prepared with only one freezing method (spontaneous freezing). In the present manuscript, we intend to detail three ways of sample freezing and to compare the results (size and number of the particles) in terms of the freezing method (e.g., the freezing rate and directionality).

The observation of the crystallization of CsCl below and above the $T_{eu}$ allowed us to present hypotheses about the mechanisms that form the particles.

**The discussion and interpretation of figure 2 is, compared to all others, very short. Specifically, the droplet-like features on figures 2a and 2b require an explanation. Rather few readers are experts in ESEM, let alone ESEM of multiphase systems!**

Thank you for your comment, the features are newly described in more detail. The following text was added:

*The black body in the upper part of the panels a-d represents sublimating ice, the grey background in the lower part of the panels represents a silicon surface of the cooling stage. Liquid brine is well visible in a form of puddles on the ice surface (white spots) and as a wide (white) borderline around the ice body. Farther from the sublimating ice, the salt already crystallized (bright white structures with a surface pattern). In the panels a and b, the crystallized CsCl salt is overexposed due to more intensive signal in comparison with the brine; to eliminate this, the sensitivity of the detector was subsequently lowered (panels c and d).*

The brine (section 3.3) must be supercooled, and indeed the droplet-like features suggest the liquid state. But is there any independent proof of the liquid nature?

We have added the following text:

*We infer the liquid state of the droplet-like features from their visual appearance and transformation to salt crystals during the observation. We did not use any other detection methods to prove their liquid state.*

**Technical corrections:**

**The surface pretreatments are not provided. They are very important for the Peltier surface and for the silicon wafer (for which the reader requires additionally how it was fixed and thermally coupled to the Peltier).**

We provided the details of the silicon surface of the Peltier stage. The following text was added to the chapter 2.1.

*Surface of the cooling stage was made from very pure, commercially available ultra-flat silicon wafer (Ted Pella, Inc.) that is usually used for the production of semiconductor components. The wafer had no additional surface treatment. It was glued to the Peltier cell with a highly thermally conductive adhesive, compatible with the low temperature and reduced gas pressure environment of the microscope.*

**Line 137: 650 Pa should not be called "ambient" pressure, which is ca. 101000 Pa.**

The "ambient pressure" refers to the pressure around the sample inside the specimen chamber of the ESEM. The term is used in the meaning of "surrounding". It shall not express the atmospheric pressure. To avoid misunderstanding, we use the term "chamber pressure" instead.

*The chamber pressure was maintained at 500 Pa.*

**Line 186: The temperature sensor and its setup are not described.**

The following text was added to chapter 2.1:

*Due to confined space and electrical interference between the temperature sensor and the detector, we are not able to directly measure the temperature on the surface of the sample during the experiment inside the ESEM chamber. Therefore, the sample temperature is inferred from the temperature of the Peltier cooler as measured by Pt1000 temperature sensor (P1K0.161.6W.A.010, Innovative sensor technology IST AG, Switzerland) installed inside the stage below the silicon pad. The actual sample temperature was validated outside the ESEM at atmospheric conditions using a thermal camera (Flir A310) and other Pt1000 temperature sensor frozen inside the sample that was placed on top of the Peltier stage. The bias between the temperature of the Peltier cooler and the actual sample temperature was no more than 2 °C.*

References:

Gitterman, K. (1937). Thermitcheskii analismorskoi vody (Thermal analysis of sea water) (*CRREL-TL-287*). Hanover, NH: Trudy Solyanoy. Laboratorii (SSSR), translation from 1971, USA Cold Regions Research and Engineering Laboratory

Thangswamy, M., Maheshwari, P., Dutta, D., Rane, V. and Pujari, P. K.: EPR Evidence of Liquid Water in Ice: An Intrinsic Property of Water or a Self-Confinement Effect?, J. Phys. Chem. A, 122(23), 5177–5189, doi:10.1021/ACS.JPCA.8B03605/SUPPL_FILE/JP8B03605_SI_001.PDF, 2018.

The list of the other references is provided in the manuscript.

---

## Author Response (AR2)

We thank again the anonymous referees for their efforts in improving the manuscript. Our responses to the comments are outlined in blue, and the modified portions of the manuscript are shown in *italics*.

**Review #1**

Many issues pointed by the reviewers were addressed in the revised manuscript. However, some unclear issues still remain in the revised manuscript. Before publication, the following issues are needed to be resolved.

Line 39: Na2SO4.10H2O

Change the terms to "Na2SO4•10H2O" Do not use "period". Use "the centered dot".

We have changed the period to the centered dot.

*The suggested mechanism of the sulphate-depletion of the SSA is mirabilite ($Na_2SO_4 \cdot 10H_2O$) precipitation...*

Line 316: silicon (1,0,0) surface

Change to silicon (1, 0, 0) surface

We have added the spaces.

*Previously, wetting the upper molecular layers of a silicon (1, 0, 0) surface...*

Line 432: mirabilite (Na2SO4•10H2O)

Because mirabilite (Na2SO4•10H2O) mentioned already in the line of 39, change to "mirabilite" or "Na2SO4•10H2O" here.

We modified the sentence as follows:

*When the ice temperature ranges below −6.4°C mirabilite starts to precipitate out from the brine...*

**Review #2**

The manuscript was very carefully corrected.

The temperature measurement is probably not ideal, but it is reproducible, and at least indirectly verified.

As for the presumably supercooled (liquid) brine, the explanation is now clear. This point had possibly been convincing to most microscopists, but not to the other readers.

"Pure" silicon wafer, without surface cleaning, might be quite contaminated, and it has an ill-defined surface chemistry, obvious from a large water contact angle (>10°), even when it is totally dust-free (as guaranteed by the manufacturer). The problem is that this change is not uniform. So it would be very good to add more details, e.g. was the wafer cut (which often entails protection by a polymer, which again changes the surface chemistry)?

We added a more detailed characterization of the wafer surface and how we handled it. We did not cut the wafer nor applied any polymeric coating; the wafer was scored by the manufacturer and we just manually (using gloves) cleaved it along the scoring into smaller tiles. However, we use the Peltier

stage also for other experiments, and although we clean the surface after each experiment (e. g. with isopropyl alcohol), we cannot exclude surface contamination due to the experimental usage.

We have modified the text followingly:

*The surface of the cooling stage was made from a pure, commercially available ultra-flat silicon wafer (P-type, boron-doped, orientation (1, 0, 0); Ted Pella, Inc.). The front surface of the wafer was polished and scored by the manufacturer; we snapped the wafer along the scoring into small tiles. We provided no additional surface treatment. The silicon tile was glued to the Peltier cell with a highly thermally conductive adhesive, compatible with the low temperature and reduced gas pressure environment of the microscope. After each experiment, we cleaned the surface of the silicon pad with isopropyl alcohol; however, surface contamination due to the experimental usage of the Peltier stage is not excluded.*